# Single-cell transcriptomics of X-ray irradiated *Drosophila* wing discs reveals heterogeneity related to cell-cycle status and cell location

Joyner Cruz, William Y Sun, Alexandra Verbeke, Iswar K Hariharan*

Department of Molecular and Cell Biology, University of California, Berkeley, Berkeley, United States

## eLife Assessment

This **important** study uses standard single-cell RNA-seq analyses combined with methods from the social sciences to reduce heterogeneity in gene expression in Drosophila imaginal wing disc cells treated with 4000 rads of ionizing radiation. The use of this methodology from social sciences is novel in Drosophila and allows them to identify a subpopulation of cells that is disproportionately responsible for much of the radiation-induced gene expression. Their **compelling** analyses reveal genes that are expressed regionally after irradiation, including ligands and transcription factors that have been associated with regeneration, as well as others whose roles in response to irradiation are unknown. This paper would be of interest to researchers in the field of DNA damage responses, regeneration, and development.

*For correspondence:
ikh@berkeley.edu

Competing interest: The authors declare that no competing interests exist.

**Abstract** Even seemingly homogeneous populations of cells can express phenotypic diversity in response to environmental changes. Thus, X-ray irradiation of tissues composed of diverse cell types can have complex outcomes. We have used single-cell RNA sequencing to study the effects of X-ray radiation on the *Drosophila* wing imaginal disc, a relatively simple tissue composed mostly of epithelial cells. Transcriptomic clustering of cells collected from the wing disc generates clusters that are mainly grouped based on proximodistal cell location. To quantify heterogeneity of gene expression among clusters, we adapted a metric used to study market concentration, the Herfindahl-Hirschman Index. Genes involved in DNA damage repair, defense against reactive oxygen species, cell cycle progression, and apoptosis are expressed relatively uniformly. In contrast, genes encoding a subset of ligands, notably cytokines that activate the JAK/STAT pathway, some transcription factors, including *Ets21C*, previously implicated in regeneration, and several signaling proteins are expressed more regionally. Though the radiation-responsive transcription factor p53 is expressed relatively uniformly in the wing disc, several regionally induced genes still require p53 function, indicating that regional and radiation-induced factors combine to regulate their expression. We also examined heterogeneity within regions using a clustering approach based on cell cycle gene expression. A subpopulation of cells, characterized by high levels of *tribbles* expression, is amplified in irradiated discs. Remarkably, this subpopulation accounts for a considerable fraction of radiation-induced gene expression, indicating that cellular responses are non-uniform even within regions. Thus, both inter-regional and intra-regional heterogeneity are important features of tissue responses to X-ray radiation.

## Introduction

Ionizing radiation (IR) is a damaging form of energy present in trace amounts in many environments and is commonly used at high doses in the form of X-rays to treat cancers. Worldwide, it is estimated that radiation therapy is an important component of treatment for more than 50% of cancers, including those that arise in the breast and lung (*Delaney et al., 2005*; *Zhu et al., 2024*). Even in the early years of radiation research, it became apparent that different types of cells varied considerably in their sensitivity to radiation. For example, lymphocytes were rapidly depleted from tissues following radiation, while tissues, such as the kidney and liver seemed far more resilient (reviewed by *McBride and Schaue, 2020*). In general, tissues that turnover rapidly are more sensitive (*McBride and Schaue, 2020*). Furthermore, within tissues, stem cells are more sensitive than their differentiated progeny (*Fabbrizi et al., 2018*), with the caveat that some types of adult stem cells, notably mesenchymal stem cells are relatively radioresistant (*Chen et al., 2006*). Even within tissues composed of relatively homogenous populations of cells, cells at particular stages of the cell cycle are more radiosensitive; mammalian cells seem most sensitive at G2/M, less so in G1 and least sensitive in late S-phase (*Pawlik and Keyomarsi, 2004*). Additionally, cells that are more hypoxic tend to be radioresistant (*Withers, 1975*; *Jain, 2014*).

Most of our mechanistic understanding of how cells react to ionizing radiation comes from genetic studies of the DNA damage response (DDR) in yeast and from biochemical studies of cultured mammalian cells (*Harper and Elledge, 2007*; *Pizzul et al., 2022*). These studies have identified the sequence of biochemical reactions that are activated by double-stranded breaks in DNA and culminate in DNA damage repair. They have also highlighted the importance of the reactive oxygen species (ROS) generated by ionizing radiation that contribute to the damage inflicted on cellular macromolecules. Importantly, cellular damage can result in the activation of mechanisms that arrest the cell cycle to enable a restoration of cellular homeostasis or, failing that, to activate pathways that promote apoptotic cell death. A key player in mammalian cells is the p53 protein which is stabilized following DNA damage and activates the transcription of genes that promote both cell-cycle arrest and apoptosis (*Levine, 2020*).

Studies in *Drosophila* have made important contributions to our understanding of the deleterious effects of IR. Indeed, the discovery that X-rays generated mutations in a dose-dependent manner was first discovered in *Drosophila* (*Muller, 1927*). Subsequent genetic studies identified loci that made flies more susceptible to the effects of DNA-damaging agents; many of these genes encode proteins now known to function in the DDR (*Sekelsky, 2017*). The wing imaginal disc, the larval precursor of the adult wings and part of the thorax (*Tripathi and Irvine, 2022*), emerged as an attractive model for studying the effects of IR on a relatively simple tissue (*Haynie and Bryant, 1977*; *James and Bryant, 1981*). The wing disc derives from a precursor population of approximately 30 cells in the embryo (*Worley et al., 2013*) and, because of cell proliferation, is composed of more than 30,000 cells by the end of the larval stages (*Martín et al., 2009*). Most of the wing disc is composed of epithelial cells and a small fraction of myoblasts which are precursors of the adult flight muscles (*Gunage et al., 2014*).

*Haynie and Bryant, 1977* irradiated larval imaginal discs and used clone-marking techniques to show that IR at a dose of 1000 rad (10 Gy) kills 40–60% of cells and irradiation with 4000 rad kills approximately 85% of cells. Despite this, compensatory proliferation allowed for the development of wings of normal size and shape. In a study using gamma irradiation (*James and Bryant, 1981*), it was shown that cell death is observed as soon as 4 hr (hr) after irradiation and continues for up to 44 hr. In parallel, there is a dramatic decrease in the number of cells undergoing mitosis within 1 hr with a resumption after 8 hr. More recent studies using flow cytometry and FUCCI have shown that cells accumulate preferentially in the G2 phase of the cell cycle after irradiation (*Ruiz-Losada et al., 2022*). Thus, as in mammalian cells, the two obvious cellular responses to IR are cell cycle arrest and apoptosis. A key difference, however, is that unlike in mammalian cells where p53 functions in both pathways, *Drosophila* p53 promotes apoptosis but does not seem to function in arresting the cell cycle (*Brodsky et al., 2000*; *Ollmann et al., 2000*). In *p53* mutants, the expression of genes that function in a wide variety of cellular responses to DNA damage is reduced, and apoptosis within 4 hr of exposure to IR does not occur. Instead, there is a delayed phase of cell death that involves multiple pathways and aneuploid cells persist in the tissue even into the pupal phase (*Akdemir et al., 2007*; *Brodsky et al., 2004*; *Brown et al., 2020*; *Wells et al., 2006*; *Wells and Johnston, 2012*; *Wichmann et al., 2006*).

Even in a relatively simple tissue, such as the wing disc, cells display considerable heterogeneity in their response to radiation. While high levels of cell death are observed in the wing pouch, the dorsal hinge shows reduced levels. The relative radioresistance of this region is dependent upon Wnt and STAT signaling (*Verghese and Su, 2016*). In more mature discs, more cell death is observed in the intervein regions of the pouch than in regions fated to generate veins (*Moon et al., 2005*). Thus, even in regions of the disc where there are no obvious morphological differences between cells, the response to radiation can differ considerably. Several studies have documented transcriptional changes at a genome-wide level in *Drosophila* embryos (*Akdemir et al., 2007*; *Brodsky et al., 2004*; *Kurtz et al., 2019*; *Lee et al., 2003*; *Ogura et al., 2009*; *Zhang et al., 2008*) and in imaginal discs *Ledru et al., 2022*; *van Bergeijk et al., 2012* following irradiation. However, because these studies have prepared RNA either from whole embryos, entire discs, or from cells from specific regions without retaining the single-cell origin of transcripts, they cannot be used to assess the heterogeneity of transcriptional responses throughout the disc.

We and others have used single-cell transcriptomics to characterize differences in the transcriptome of cells from different parts of the wing disc (*Bageritz et al., 2019*; *Deng et al., 2019*; *Everetts et al., 2021*; *Zappia et al., 2020*). Since the dominant sources of transcriptional variability between cells, or stratifying factors, reflect differences in cell location along the proximodistal (PD) axis, cells from different regions of the disc can easily be identified. Additionally, we and others have characterized transcriptional changes at the single-cell level after ablation of the wing pouch using the tumor necrosis factor (TNF) ortholog *eiger* (*egr*) (*Floc'hlay et al., 2023*; *Worley et al., 2022*) and we have previously identified a pathway downstream of the Ets21C transcription factor that functions during regeneration but not during wing disc development (*Worley et al., 2022*). These studies provide a foundation for characterizing transcriptional responses of the wing disc to IR and to examine the level of transcriptional heterogeneity between cells.

Here, we present a comparison, at the single-cell level, of the transcriptomes of unirradiated and irradiated wing discs from late third-instar larvae. Our studies reveal heterogeneity at two different levels – between territories in the disc and between cells in individual territories. We show that regional heterogeneity is more a feature of some classes of genes over others and use quantitative approaches to investigate heterogeneity more generally.

## Results

### X-ray irradiation induces widespread DNA damage, but apoptosis and cell cycle arrest occur nonuniformly

The wing disc consists of three major regions that span its PD axis (*Figure 1A*): The notum, which generates most of the adult thorax, the hinge, which develops into an adult structure of the same name that connects the wing blade to the thorax, and the pouch, which develops into the wing blade (*Klein, 2001*; *Ayala-Camargo et al., 2013*). We first characterized the primary effects of X-rays on these regions. In order to confirm that X-rays induce DNA damage in each of the major PD regions, we used immunofluorescence to visualize phosphorylated histone H2Av (p-H2Av), a modified histone variant generated after DNA damage that serves as an early mark of double-strand break repair (*Madigan et al., 2002*).

We found that in unirradiated discs, p-H2Av fluorescence was present in relatively low levels throughout the tissue (*Figure 1B*). Thirty minutes (min) after irradiation, a time point conducive to capturing an initial response to DNA damage (*Madigan et al., 2002*; *Porrazzo et al., 2022*; *Apostolou et al., 2025*), we observed an increase in p-H2Av signal in each of the PD regions, with some inter-regional variability (*Figure 1C*). To test for X-ray induced cell death, we performed immunofluorescence (IF) staining using an antibody that targets cleaved Dcp-1, an effector caspase that is active during apoptosis (*Song et al., 1997*). Unirradiated discs showed low levels of Dcp-1 signal, likely associated with normal development (*Figure 1D*). Four hours (hr) after X-ray exposure, discs showed a marked increase in Dcp-1 signal in each of the major regions of the disc (*Figure 1E*). When compared to p-H2Av fluorescence, anti-DCP1 signal is less uniform. High levels of apoptosis were observed in the wing pouch, but cells along the dorsoventral boundary were spared. Additionally, apoptosis is reduced in portions of the wing hinge, as has been described previously (*Verghese and Su, 2016*).

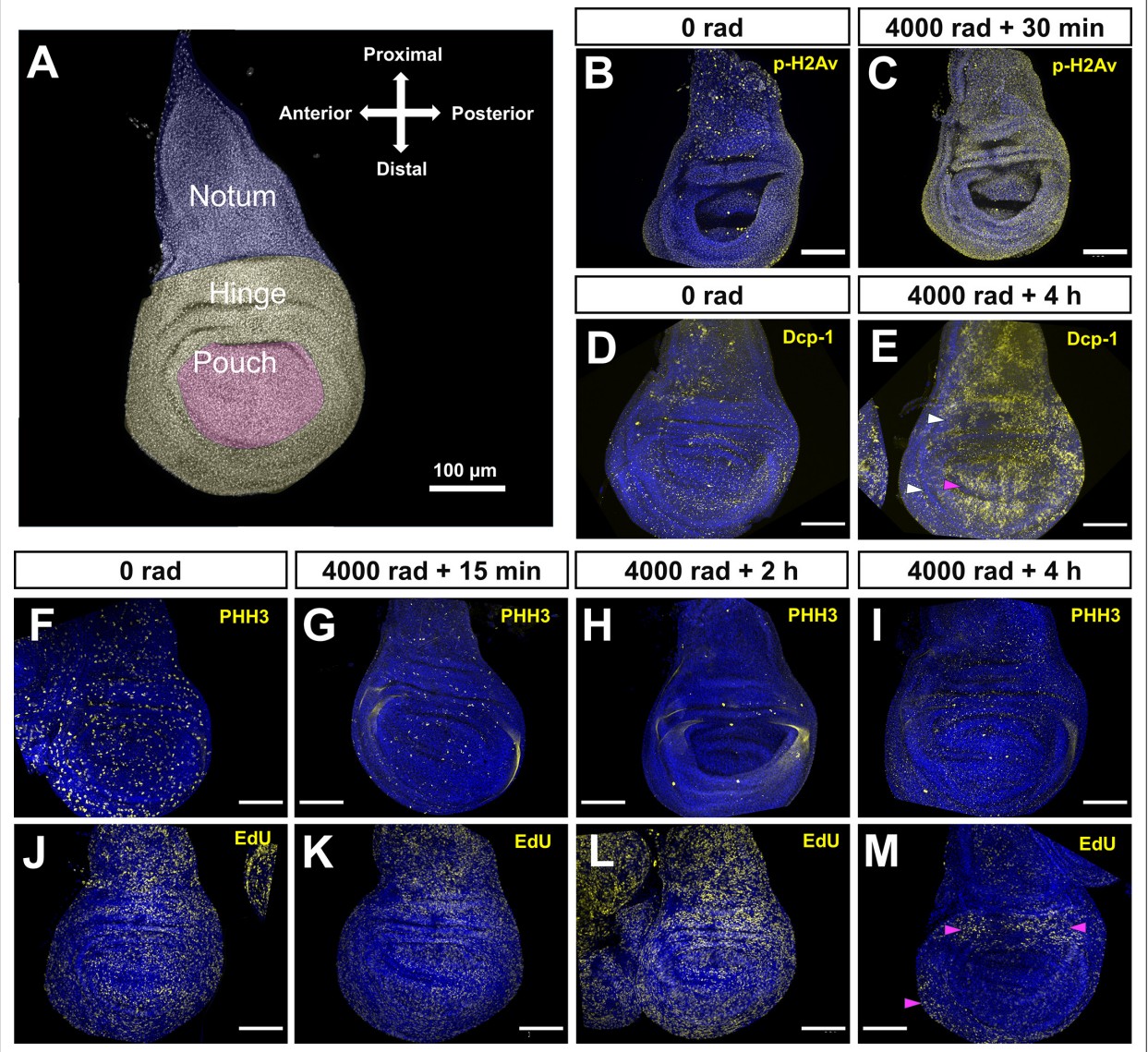

**Figure 1.** Effects of X-ray irradiation on DNA damage, apoptosis, and cell cycle progression. (**A**) Cartoon overlay of wing disc showing proximodistal (PD) regions. (**B, C**) Immunofluorescence (IF) of p-H2Av at 0 rad (**B**) and 4000 rad of irradiation, 30 min after exposure (**C**). (**D, E**) IF of cleaved Dcp-1 at 0 rad (**D**) and 4000 rad 4 hr after exposure (**E**). Magenta arrowhead points to the DV boundary and white arrowheads point to regions of the hinge with less Dcp-1 signal. (**F–I**) IF of PHH3 at 0 rad (**F**) and 4000 rad 15 min (**G**), 2 h (**H**), and 4 hr (**I**) after exposure. (**J–M**) EdU incorporation at 0 rad (**J**) and 4000 rad 15 min (**K**), 2 h (**L**), and (**M**) 4 hr after exposure. Magenta arrowheads point to regions with high EdU signal at 4 hr after irradiation. All scale bars are 100 μm. For B–M, DAPI is in blue.

To examine the proliferative effects of X-ray exposure, we used IF against phosphohistone H3 (PHH3), a modified H3 histone state associated with chromatin condensation during mitosis (*Hendzel et al., 1997*). PHH3-stained chromosomes were observed across all regions of unirradiated discs (*Figure 1F*). In irradiated discs, we observed a global reduction in the number of PHH3-stained chromosomes as soon as 15 min after exposure (*Figure 1G*) with greater reduction at 2 and 4 hr after exposure (*Figure 1H–I*). This is consistent with prior observations that cells accumulate in G2 following IR exposure and fail to enter mitosis (*Song et al., 2004*; *Yuan et al., 2018*; *Ruiz-Losada et al., 2022*). As an additional measure of proliferation, we visualized DNA synthesis using EdU labeling, where positive labeling is associated with an active S-Phase (*Salic and Mitchison, 2008*). Like PHH3 staining, EdU labeling was observed in all regions of the tissue in unirradiated discs (*Figure 1J*). Up to 2 hr after irradiation, the distribution of EdU labeling resembled that of unirradiated discs (*Figure 1K–L*), while at 4 hr after exposure, there was a consistent reduction in EdU labeling in the notum and

pouch regions of the tissue, while the hinge was partially spared from this reduction (*Figure 1M*). The persistence of EdU incorporation in the hinge could either indicate that a proportion of cells in this region continue to enter S-phase or, alternatively, that there are higher levels of DNA synthesis related to DNA repair in this region. Taken together with previous studies, our observations indicate that different regions of the disc show differences in response to radiation both in terms of parameters related to cell cycle progression and to apoptosis.

## Cells of unirradiated and irradiated wing discs show similar patterns of expression of many regionally expressed genes

Since we have observed differences in cell death and DNA synthesis across the disc, it is likely that there are transcriptomic differences between regions. To explore changes in gene expression on a genome-wide basis across the disc, we used single-cell RNA sequencing (scRNA-seq). Previous scRNA-seq studies of the disc have shown that genes with patterns of expression restricted to different regions of the PD axis can be used as markers to estimate the approximate anatomical location of cells (*Bageritz et al., 2019*; *Deng et al., 2019*; *Everetts et al., 2021*; *Zappia et al., 2020*). These include *zfh2* expressed in the hinge (*Figure 2A and A"*) and *nubbin* (*nub*) expressed in the pouch (*Figure 2A' and A"*). In irradiated discs, we found no major changes in the general expression pattern of either gene (*Figure 2B–B"*). This suggests that these two marker genes can be used to accurately identify hinge and pouch cells in both conditions.

To examine gene expression on a genome-wide scale, we performed scRNA-seq on wing discs collected from late third instar larvae 4 hr after exposure to 4000 rad X-rays, and from larvae of the same stage that were unexposed to irradiation. For both samples, two replicates were collected. To generate datasets containing high-quality cells, several filtering steps were applied. In brief, cells that were positively stained with propidium iodide, indicating a compromised cell membrane, were removed via FACS. After sequencing, cells within each dataset that had low numbers of captured genes were removed from further analysis. While cells from all major PD regions of the disc were present in our analysis, our filtering approach presumably removed cells that were furthest along the pathway to cell death. Therefore, filtering our 4000 rad samples may have removed those cells which were most damaged by X-rays or were most sensitive to them resulting in an inherent bias in the type of cells that were analyzed. For analysis, all four datasets were integrated using Seurat v5's anchor-based canonical correlation analysis (CCA) integration process (see Methods for details). Cells were grouped into clusters in the integrated dataset using the Louvain algorithm (default in Seurat v5) with a resolution parameter of 2, resulting in 35 clusters (*Figure 2C*).

Each dataset was individually stratified across its PD axis, as well as when integrated, as indicated by a separation of PD markers between clusters (*Figure 2D–H*). Clusters were classified as belonging to one of seven broad PD regions based on marker expression: the pouch, hinge-pouch, hinge, hinge-notum, notum, PE, and myoblasts. For the pouch, hinge-pouch, and hinge: *nub+, zfh2-* clusters were annotated as pouch, *nub+, zfh2 + +* clusters as hinge-pouch, and *nub-, zfh2 + +* as hinge. For the hinge-notum and notum: *tup+* (or *eyg+*, *Figure 2—figure supplement 1A*), *zfh2- clusters* were annotated as notum, and *tup+* (or *eyg+*), *zfh2 +* as hinge-notum. The one *twi +* cluster was annotated as myoblasts (*Figure 2—figure supplement 1B*). The classification of clusters into broad regions is imperfect, as there are some examples of clusters expressing moderate levels of individual marker genes belonging to the region that neighbors the one that they were classified into (e.g. *zfh2* is expressed at moderate levels in some cells belonging to pouch cluster 2, P2).

To identify unique cluster markers, we performed differential gene expression analysis on each individual cluster compared to all other clusters and compared to all other clusters within the same broad region (*Figure 2—figure supplement 1C*, *Supplementary file 1*). Unique markers indicated that some clusters likely correspond to contiguous places in the wing disc with regionally restricted gene expression of one or more genes, e.g., pouch cluster 5 (P5), marked by *Optix*, a gene restricted in expression to a small region of the anterior pouch (*Seimiya and Gehring, 2000*). Other clusters likely correspond to non-contiguous regions in the wing disc with restricted gene expression of one or more genes, e.g., Pouch cluster 7 (P7), marked by *Doc1*, a gene expressed in two separate locations of the dorsal and ventral pouch (*Butler et al., 2003*) and by previous observations that cells from the anterior and posterior compartments can both be found in several clusters (*Bageritz et al., 2019*; *Deng et al., 2019*; *Everetts et al., 2021*; *Zappia et al., 2020*). In two cases, clusters did not correspond to specific

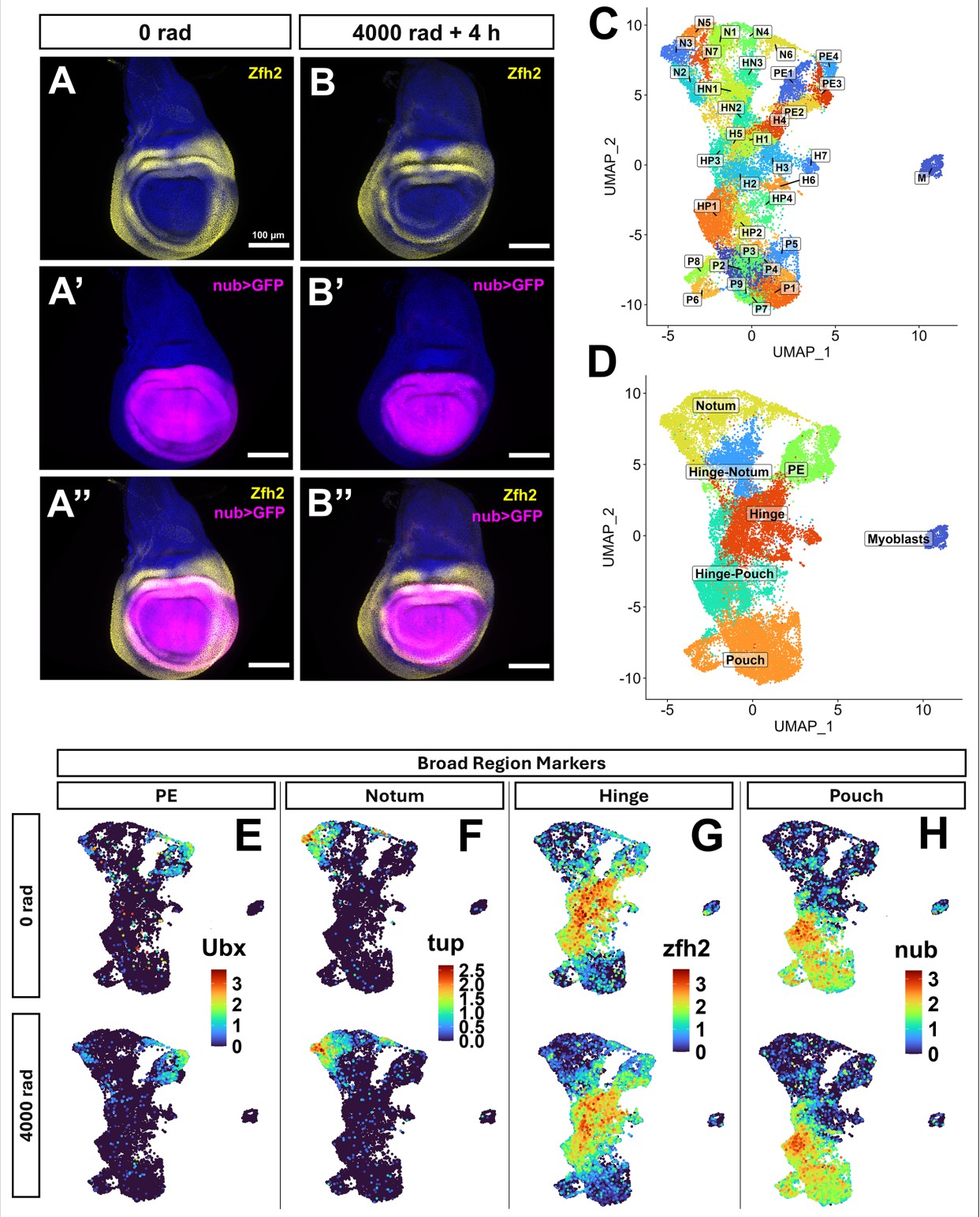

**Figure 2.** Territories of the wing disc shown by immunofluorescence and UMAP plots. Immunofluorescence (IF) of Zfh2 (**A, B**), GFP expression in w-, nub-gal4, uas-GFP; fly wing discs (**A', B'**), and merged images of both (**A''**, B") at 0 rad (**A–A''**) or 4 hr after exposure to 4000 rad (**B–B''**). For A–B', DAPI is in blue. All scale bars are 100 µm. (**C**) UMAP showing 35 cluster annotations with cluster annotations based on proximodistal (PD) region. (**D**) UMAP showing broad epithelial region annotations. (**E–H**) UMAP plots showing the expression of markers used to annotate specific epithelial regions at 0 rad (top row) and 4000 rad (bottom row); *Ubx* for PE (**E**), *tup* for notum (**F**), *zfh2* for hinge (**G**), *nub* for pouch (**B**). Combinations were used to determine

*Figure 2 continued on next page*

*Figure 2 continued*

hinge-notum and hinge-pouch regions. Plots for *eyg* (also used for the notum) and *twi* (used for the myoblasts) are shown in **Figure 2—figure supplement 1**.

The online version of this article includes the following figure supplement(s) for figure 2:

**Figure supplement 1.** UMAP plots of top cluster markers by region.

regions of restricted gene expression: pouch cluster 9 (P9), which was enriched for cells with S-phase markers, and pouch cluster 3 (P3), which was enriched in irradiated cells expressing the cell cycle gene *tribbles (trbl)* (later discussed at length). In sum, we interpret cluster identity to be primarily associated with genes of restricted, though not necessarily contiguous, regional patterns of expression in the disc with some additional contributions from other parameters, such as cell-cycle state.

## X-ray-induced expression of genes involved in apoptosis, DDR, and reaction to ROS is relatively homogeneous across the disc

For an initial broad survey of changes in gene expression after irradiation, we compared the gene expression of all cells of the irradiated condition to all cells of the unirradiated condition. We applied lenient filters, retaining genes that were present in ≥1% of cells in either condition, had an absolute $log_2FC$ ≥0.1 (FC refers to fold change), corresponding to an approximately 7% increase or decrease in abundance, and an adjusted *p*-value <0.05, resulting in 3767 genes with increased expression in the irradiated condition and 1122 genes with decreased expression. Among genes with increased expression were many of those described in previous genome-wide analyses on embryos and wing discs, including genes involved in apoptotic regulation (e.g. *rpr, hid, Corp, egr)* and genes involved in DDR and repair (e.g. *Irbp18, Xrp1, p53, Ku80, Irbp, mre11, rad50)* (**Akdemir et al., 2007**; **Brodsky et al., 2004**; **van Bergeijk et al., 2012**). Similarly, among genes with decreased expression were many described in previous studies of irradiated wing discs (**van Bergeijk et al., 2012**). These included genes involved in DNA replication (e.g. *PCNA, dup, Mcm3, geminin, PolE2*). For all changes, see **Supplementary file 2**.

There are currently few methods used routinely to assess the heterogeneity of gene expression across cell clusters. There are a variety of ways that heterogeneity is quantified in economics and the social sciences (reviewed in **Steele et al., 2022**). To quantify the heterogeneity of X-ray-induced genes across cell clusters, we implemented a variation of the formula used in the Herfindahl-Hirschman Index (HHI), a measure of market concentration used in economics (**Herfindahl, 1950**; **Hirschman, 1945**) (here used to measure the 'concentration' of gene expression across clusters). If the share of total expression in each cluster is calculated and if those numbers are simply all added, the overall total would be 1. If, however, the share of expression in each cluster is squared and then those numbers are added, the total would depend upon the distribution of expression between the clusters. If expression was distributed evenly among N clusters, then each cluster would have 1/N of the total expression. In that case, the HHI would be 1/N which is the lowest possible value. In contrast, if all expression was concentrated in one cluster, then the HHI would be 1, which would be the highest possible value and which would be indicative of maximal heterogeneity between clusters. To calculate the HHI score for a given gene, the average gene expression under the irradiated condition for each of the 35 clusters was individually divided by the sum of all average expressions across clusters, squared, and then added together (**Figure 3A**, formula on top). Thus, in the case of 35 clusters, the minimum possible score is 0.0286, and the maximum possible score is 1. We consider HHI scoring to be a useful heuristic method for ranking expression concentration in clustered data. HHI applied to clusters here, which are primarily defined by genes with spatially restricted expression, allows us to rank genes in terms of their spatial heterogeneity.

To see if there were differences in expression concentration between classes of genes that may be important for X-ray response, we applied HHI using the 35 clusters to 521 genes enriched at 4000 rad that belonged to one of nine categories: apoptosis, DDR, response to ROS, cell cycle regulation, transcription factors (TFs), phosphatases, kinases, ligands, and receptors (**Figure 3A**; see **Supplementary file 3** for complete list of genes used before filtering to 521 genes which are shown in **Supplementary file 5**. For the gene filtering approach, see methods). The mean HHI for the 521 genes was 0.037 with a minimum value of 0.029 (*eff*) and a maximum value of 0.25 (*tup*), the theoretical range

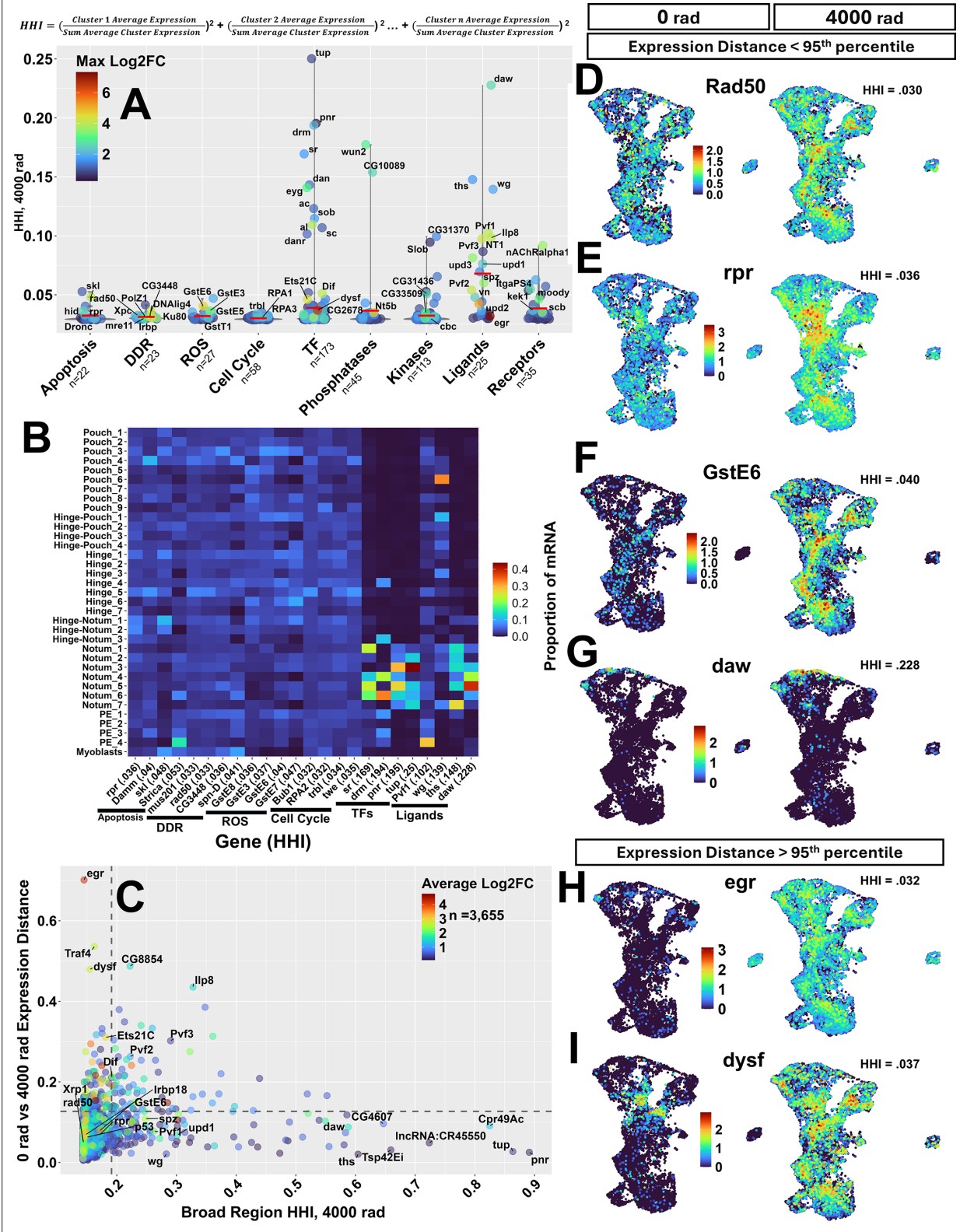

**Figure 3.** Heterogeneity of gene expression across the wing disc following radiation exposure. (**A**) 521 genes across nine functional categories plotted as points by their HHI score at 4000 rad. Point color represents the log₂FC between conditions in the cluster that has the highest log₂FC. The horizontal red bar represents the mean HHI score for the genes of that category. Genes with max log₂FC >2 or HHI >0.075 were labeled where space permitted. The equation used to calculate the Herfindahl-Hirschman Index (HHI) is shown above the panel. (**B**) Heat map of four highest HHI scoring genes in

*Figure 3 continued on next page*

*Figure 3 continued*

apoptosis, DNA damage response (DDR), reactive oxygen species (ROS), cell cycle categories (low HHI scoring categories), and transcription factor (TF) and ligand categories (categories with high HHI scoring genes). Box colors represent the proportion of mRNA found in each subregion relative to the sum of all mRNA found in all clusters (the values used to calculate HHI). (**C**) X-ray induced genes plotted by their HHI score (calculated on seven broad regions, not the 35 clusters) and the Euclidean distance in gene expression in seven-dimensional space at 0 rad vs 4000 rad (calculated using the seven broad regions as described in the text). Point color represents the $log_2$FC of each gene when comparing all cells from the 4000 rad condition to all cells of the 0 rad condition. The dotted lines are drawn at the 95th percentile for each of the two parameters. (**D**–**G**) Gene expression UMAPs of example genes with <95th percentile Euclidean distance score belonging to the DDR (**D**), apoptosis (**E**), ROS (**F**), and ligand (**G**) groups. (**H**–**I**) Gene expression UMAPs of two genes with large differences in expression pattern between 0 rad and 4000 rad identified by a top 5% Euclidean distance. Genes in (**D**–**I**) are arranged from high (top) to low (bottom) HHI score.

The online version of this article includes the following figure supplement(s) for figure 3:

**Figure supplement 1.** Histogram of cluster Herfindahl-Hirschman Index (HHI) scores, broad region HHI scores, and Euclidean expression distance scores.

**Figure supplement 2.** Proportion mRNA heatmaps of ligands and transcription factors.

**Figure supplement 3.** Proportion mRNA heatmaps transcription factors continued.

**Figure supplement 4.** Proportion mRNA heatmaps of apoptosis, reactive oxygen species (ROS), cell cycle, and DNA damage response (DDR) genes.

**Figure supplement 5.** Proportion mRNA heatmaps of kinase, phosphatase, and receptor genes.

**Figure supplement 6.** Examples of genes spanning a range of Euclidean expression distance scores.

being between 0.0286 (least concentrated) and 1 (most concentrated). For HHI score distributions of all 3767 X-ray induced genes, see *Figure 3—figure supplement 1A*. The classes of genes that have been studied most intensively in the context of cellular responses to radiation (apoptosis, DDR, ROS, cell cycle) are all expressed relatively homogeneously with HHI scores less than 0.06. We found that most genes with the highest HHI scores tended to encode either ligands or transcription factors, indicating their concentrated expression in relatively few clusters. A small number of genes that encoded phosphatases, kinases, or receptors were also expressed relatively heterogeneously. Another way to visualize the relative heterogeneity of expression is to show the proportion of mRNA of each gene in each of the 35 clusters. Genes with the highest HHI from six different classes are shown demonstrating greater spatial expression concentration of the TFs and ligands than the other categories. (*Figure 3B*; *Figure 3—figure supplements 2–4*). Likewise, expression UMAPs of low HHI genes (*Figure 3D–F*) showed less concentrated patterns of expression when compared to high-scoring genes (*Figure 3G*).

Of the 27 genes with the highest HHI scores (95th+ percentile of 521 genes, HHI ≥0.066, n=27), 10 were ligands, including JAK/STAT pathway activators *upd1* (HHI = 0.076), concentrated in clusters of the hinge, hinge-notum, and PE, and *upd3* (HHI = 0.081) concentrated in clusters of the pouch, hinge-pouch, and hinge. Expression of the VEGF/PDGF orthologs *Pvf1* (HHI = 0.102) and *Pvf3* (HHI = 0.097), which bind to the Pvr receptor tyrosine kinase, is most concentrated in clusters of the PE. Expression of *Ilp8* (HHI = 0.098), which encodes a member of the insulin/relaxin family and regulates systemic responses to disc injury is concentrated in the hinge-pouch. Each of these genes is known to be upregulated in response to various types of disc injury (*Blanco et al., 2010*; *Floc'hlay et al., 2023*; *Katsuyama et al., 2015*; *Worley et al., 2022*). Given that radiation induces damage uniformly, the relatively localized expression of these genes was unexpected. The four transcription factors with the highest HHI scores, *tailup* (*tup*) (HHI = 0.250), *pannier* (*pnr*) (HHI = 0.195), *drumstick* (*drm*) (HHI = 0.194), and *stripe* (*sr*) (HHI = 0.169), are all known to function in cell fate determination. For HHI scores on all genes included in this analysis, see *Figure 3—figure supplements 2–5* and *Supplementary file 5*.

When examining gene expression using HHI, we noticed that many X-ray induced genes with concentrated expression at 4000 rad had a lower, but similar pattern of expression at 0 rad (e.g. *daw*, *Figure 3G*, and *upd1*, *Figure 4B and C*). To assess whether this property applied to many genes, we implemented a measure of the difference in the pattern of gene expression between conditions on the 3767 X-ray-induced genes. To reduce noise, we further removed genes not found in at least 1% of cells in both conditions, resulting in 3655 genes. For each gene, we calculated the proportion of mRNA found in each of the seven PD regions (PE, notum, hinge-notum, hinge, hinge-pouch, pouch, and myoblasts) under each condition. We chose to use the seven PD regions rather than the 35 clusters to determine large, coarse changes in gene expression across the PD axis. Thus, each gene

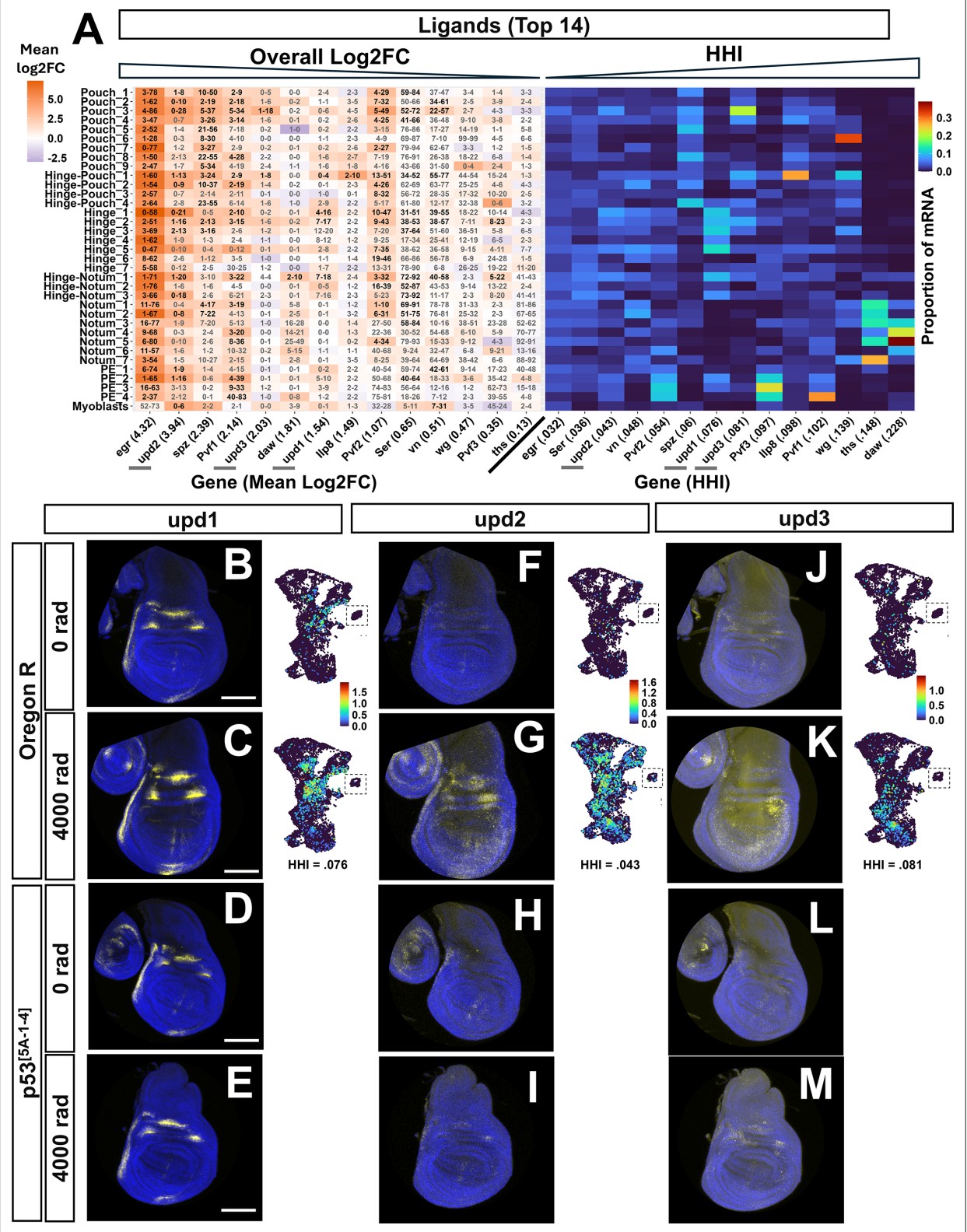

**Figure 4.** X-ray-induced expression of genes encoding ligands. (**A**) Heatmaps showing $\log_2$FC between conditions for each cluster (left) and proportion of total mRNA in each cluster at 4000 rad (right) of the top 14 ligands ranked by max $\log_2$FC in any cluster. A Left: Boxes are colored based on the average $\log_2$FC from 0 rad to 4000 rad for each cluster. The left number in each box is the percentage of cells in the cluster expressing at least one transcript at 0 rad; the right number is the same at 4000 rad. Dark black numbers indicate that the change is statistically significant in that cluster

*Figure 4 continued on next page*

*Figure 4 continued*

(adjusted *p*-value <0.05, Wilcoxon Rank Sum Test, Bonferroni correction), while light gray numbers indicate that it is not (adjusted *p*-value ≥0.05). Genes are sorted from left to right in descending order of the mean $\log_2$FC when comparing all cells of 0 rad to all cells of 4000 rad ('overall mean $\log_2$FC'); the overall mean $\log_2$FC value is in parentheses next to gene names (all significant). A Right: Boxes are colored based on the proportion of mRNA expressed in that cluster vs the total amount of mRNA expressed in all clusters. Genes are sorted from left to right in ascending order of Herfindahl-Hirschman Index (HHI) score. HHI score is noted in parentheses next to gene name. Genes shown in panels (**B–G**) are underlined with gray bars in (**A**). (**B–M**) In each panel: hybridization chain reaction (HCR) signal of *upd1, upd2, and upd3* are in yellow and DAPI in blue at 0 rad (**B, D, F, H, J, L**) or 4000 rad (**C, E, G, I, K, M**). Wild-type discs (**B, C, F, G, J, K**) are compared to p53 mutant discs (**D, E, H, I, L, M**). To the right of each wild-type HCR image is an expression UMAP of each gene in each condition. The myoblast cluster was cropped into UMAP images for space, indicated by the dotted box around them.

The online version of this article includes the following figure supplement(s) for figure 4:

**Figure supplement 1.** Toll and PDGF/VEGF ligands and receptors.

**Figure supplement 2.** JAK/STAT receptors and tumor necrosis factor (TNF) ligands and receptors.

could be represented as a point in seven-dimensional space in each of the two conditions. We then calculated the Euclidean distance ('expression distance') between these points. Gene scores for this measure ranged from 0 to 0.7, with higher scores indicating a greater difference in the overall pattern of expression between conditions (*Figure 3C*).

Scores formed a right-skewed distribution, with most genes (>95%) scoring <0.13 (*Figure 3—figure supplement 1C*). We found that genes below this threshold, when inspected using expression UMAPs (*Figure 3—figure supplement 6C*), displayed some level of conserved expression patterns between conditions, with lower scoring genes having more conserved patterns of expression. Examples from this category include the DNA repair gene *Rad50* (*Figure 2D*), the pro-apoptotic gene *reaper* (*rpr*) (*Figure 2E*), *GstE6* which responds to oxidative stress (*Figure 2F*) and activin-like ligand *dawdle* (*daw*) (*Figure 2G*). The genes with the most marked changes in expression pattern, accounting for the top 5% of genes with the highest Euclidean distance between the two conditions, included the TNF ortholog *eiger* (*egr*) and the transcription factor *dysfusion* (*dysf*) (*Figure 3H and I*) whose expression changes from being relatively localized at 0 rad to being widespread at 4000 rad. To examine the relationship between expression distance and expression concentration in the seven broad regions, we also calculated the HHI score of genes using the proportion of mRNA in each broad region ('BR-HHI;' for BR-HHI score distributions, see *Figure 3—figure supplement 1B*) and plotted them against expression distance (*Figure 3C*). Genes with high BR-HHI scores in the 4000 rad condition (those whose expression is most heterogeneous among these regions) tended to have relatively low expression distance scores (e.g. *daw, tup, pnr),* indicating that they shared a similar pattern of expression at 0 rad. The reason for this similarity of expression pattern under the two conditions is not obvious. One possible explanation is that genes that are transcriptionally active under unirradiated conditions in some regions of the disc have more accessible chromatin configurations in those cells and are, therefore, more easily induced in those same cells following irradiation. Another is that induction of these genes occurs by the combined action of region-specific transcription factors and those that are induced by radiation.

## The TNF ortholog *eiger* is expressed relatively homogeneously in all PD regions, while ligands of the Toll, PDGF/VEGF, and JAK/STAT pathway show regionally induced expression

To examine differences in the expression of genes with high HHI scores more closely, we focused on the 14 ligands with the highest FC at the cluster level after X-ray exposure. Among the top 14 ligands were genes belonging to the TNF pathway, the Toll pathway, PDGF/VEGF-related pathway, and the JAK/STAT pathway.

The most highly induced and uniformly expressed ligand was the tumor necrosis factor (TNF) pathway activator *eiger* (*egr*), increasing more than 16-fold overall after irradiation (*Figure 4A* left) with an HHI score of 0.32 (*Figure 4A* right). In the 4000 rad condition, *egr* had the lowest HHI score among ligands, with expression in all major regions of the PD axis (*Figure 4A* right). The two genes encoding *egr* receptors, *grnd,* and *wgn,* are expressed in the disc in both conditions. *grnd* had an overall positive fold change after X-ray exposure and is expressed relatively uniformly across clusters (HHI = 0.030), while *wgn* has an overall negative fold change and is concentrated in the pouch

and hinge-pouch (*Figure 4—figure supplement 2B*). Eiger activates the JNK pathway, which was previously shown to function in promoting p53-independent cell death (*McNamee and Brodsky, 2009*).

The Toll pathway is well known as a branch of innate immunity in *Drosophila* that is also important in wound healing (*Capilla et al., 2017*; *Carvalho et al., 2014*). Two of six Toll pathway ligands were enriched after X-ray exposure, with *spz* (HHI = 0.060) being among the top 14 induced ligands, the second ligand being *spz3* (HHI = 0.034). For *spz*, the clusters of highest FC belonged to the pouch and hinge-pouch (*Figure 4A* left), with expression in the 4000 rad condition being concentrated in these regions (*Figure 4A* right). Seven of nine Toll receptors in *Drosophila* were also detected in the wing disc. Toll receptor genes *18 w, Tl, Toll-7,* and *Tollo* were detected at high levels, *Toll-9* at low levels, and *Tehao and Toll-6* at barely detectable levels. *18 w, Tl, Toll-7,* and *Tollo* were expressed in varying patterns across clusters and had overall reduced expression after X-ray exposure (For Toll receptors and their ligands, see *Figure 4—figure supplement 1A, B*).

PDGF/VEGF-related signaling in *Drosophila* is known to function in cell migration and wound closure (*Tsai et al., 2022*). All three *Drosophila* PDGF/VEGF-related ligands, *Pvf1, Pvf2,* and *Pvf3,* were among the top 14 upregulated after X-ray exposure, with *Pvf1* and *Pvf2* having the highest overall FC ($\log_2$FC 2.14 and 1.07, respectively, *Figure 5A* left). All three ligands showed a relatively high level of expression at 0 rad in the PE (*Figure 4—figure supplement 1D*) which was maintained at 4000 rad (4 A right). The three Pvf genes, *Pvf1, Pvf2,* and *Pvf3* had HHI scores of 0.102, 0.054, and 0.097, which reflect their relatively heterogeneous expression (*Figure 4A* right, *Figure 4—figure supplement 1D*). The sole PDGF/VEGF receptor in *Drosophila*, *Pvr* (HHI = 0.034), is expressed relatively uniformly across all clusters in both conditions, being slightly elevated in the 4000 rad condition, consistent with the possibility that all major regions of the disc are capable of responding to Pvf ligands.

In *Drosophila,* the JAK/STAT pathway is required for normal development but is also known to be important for regeneration of the wing disc (*Herrera and Bach, 2019*; *Katsuyama et al., 2015*). All three *Drosophila* JAK/STAT ligands, *upd1* (*Harrison et al., 1998*), *upd2* (*Rajan and Perrimon, 2012*), and *upd3* (*Romão et al., 2021*), were among the top 14 most induced ligands in scRNA-seq: *upd1* was primarily induced in the hinge and hinge-notum after irradiation, with concentrated expression in these regions in the 4000 rad condition (HHI = 0.076), that matched a similar, but lesser pattern of expression at 0 rad (*Figure 4A* and 4B–4 C right). *upd2* and *upd3* were expressed at low levels across all clusters at 0 rad (*Figure 4F and J*). *upd2* was induced at varying levels in all PD regions of the disc epithelium as well as the myoblasts (HHI = 0.43), and *upd3* was primarily induced in the pouch and hinge-pouch after X-ray exposure (HHI = 0.81) (*Figure 4G–K* right). To validate these findings, we performed RNA in situ hybridization using hybridization chain reaction (HCR) on upd gene transcripts in the 0 and 4000 rad conditions. *upd1* displayed a previously characterized pattern of restricted expression in the hinge and hinge-notum (*Johnstone et al., 2013*) that was enhanced after irradiation, which was consistent with its expression UMAP (*Figure 4B and C*). *upd2* was most highly expressed in the hinge/pouch of the wing disc with low expression in the notum (*Figure 4F–G*). *upd3* was most strongly expressed in the pouch and hinge pouch with low/undetectable expression in the hinge, consistent with its expression UMAP (*Figure 4K*). The sole JAK/STAT receptor in *Drosophila*, *dome*, is expressed in all clusters at 0 rad with a slight increase in expression at 4000 rad (HHI = 0.031) (*Figure 4—figure supplement 2A*). Thus, for both the Pvf-family ligands and the upd-family ligands, expression of the ligands is relatively heterogeneous when compared to the receptors whose expression is more uniform.

The expression of *upd2* and *upd3* in *RasV12* tumors has been shown to be dependent on p53 and confer tumor radiation resistance in *Drosophila* (*Dong et al., 2021*). We, therefore, sought to determine if radiation-induced expression of the upd genes required *p53* in normally developing irradiated wing discs. In wing discs carrying a near-complete deletion of *p53* (*Xie and Golic, 2004*), all three upd genes showed little to no induction after X-ray exposure (*Figure 4H, I, L and M*), unlike their wild-type counterparts. However, *upd1* in *p53* mutants had a normal pattern of expression in regions of the hinge before irradiation (*Figure 4D*), similar to wild-type wing discs. Together, these results indicate that X-ray induction of *upd1, upd2,* and *upd3* expression requires *p53*, while the normal developmental pattern of *upd1* expression is independent of *p53*.

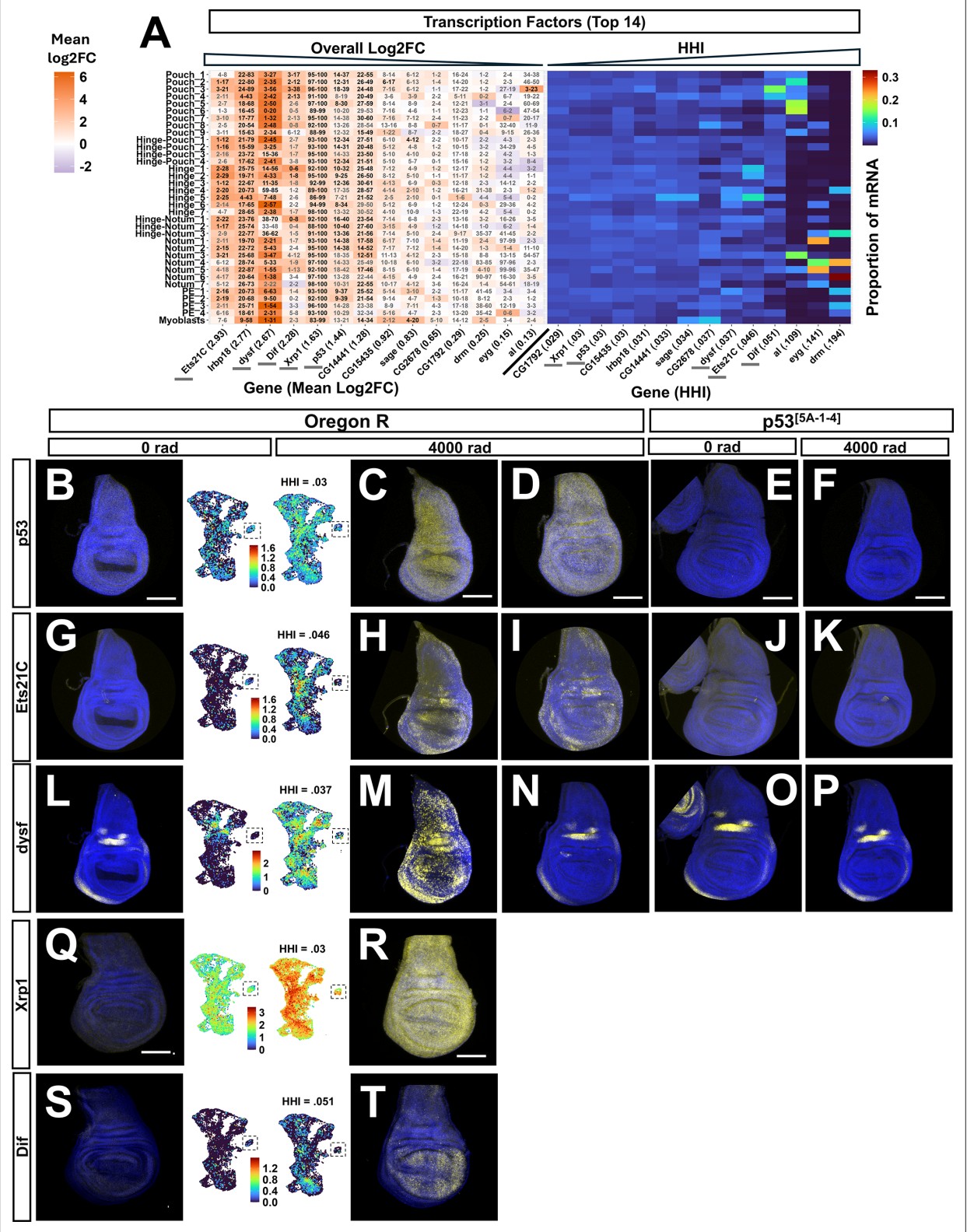

**Figure 5.** X-ray induced expression of transcription factors. (**A**) Heatmaps showing log₂FC between conditions for each cluster (left) and proportion of total mRNA in each cluster at 4000 rad (right) of the top 14 transcription factors (TFs) ranked by max log₂FC in any cluster. Generated in the same way as *Figure 4A*. Genes shown in panels (**B–T**) are underlined with gray bars in (**A**). (**B, G, L, Q, S**) hybridization chain reaction (HCR) of *p53, Ets21C, dysf, Xrp1*, and *Dif* at 0 rad in Oregon R wing discs. (**C, H, M, R, T**) HCR of the same genes at 4000 rad in Oregon R wing discs. For B–T, HCR signal

*Figure 5 continued on next page*

*Figure 5 continued*

is in yellow and DAPI in blue. To the right of 0 rad and left of 4000 rad HCR image is an expression UMAP of each gene in each condition. (**D, I, N**) Alternative irradiated wing disc showing non-induction of *dysf* after X-ray exposure. The myoblasts were cropped into the UMAP images for space, indicated by the dotted box around them. (**E, J, O**) HCR of *p53, Ets21C,* and *dysf* in *p53[5 A-1-4]* mutant wing discs at 0 rad. (**F, K, P**) HCR signal of the same genes at 4000 rad in p53[5 A-1-4] mutant wing discs.

The online version of this article includes the following figure supplement(s) for figure 5:

**Figure supplement 1.** Expression UMAP of Toll transcription factor (TF), high magnification images of *dysf* hybridization chain reaction (HCR), and increased intensity images of *Xrp1* and *p53*.

## Transcription factors Dif and Ets21C are expressed heterogeneously in irradiated wing discs, while expression of p53, Irbp18, and Xrp1 is relatively homogenous

Similarly to ligands, we focused our attention on the TFs with the maximum FC in any cluster after X-ray exposure (*Figure 5A*). Among these 14 TFs were those involved in DDR and were previously shown to be upregulated after IR, including *p53, Irbp18, and Xrp1* (*Akdemir et al., 2007*; *Brodsky et al., 2004*; *Khan and Baker, 2022*), and *dysf*, a TF important for the development of several structures, including the trachea (*Jiang and Crews, 2003*), but yet to be characterized in the context of X-ray response. Also included were genes involved in regeneration (*Ets21C*), immunity (*Dif*), and several predicted TFs with unknown functions (*CG14441, CG15435, CG2678, CG1729*).

*Xrp1* and *Irbp18* are two transcription factors important for DDR that form a heterodimeric unit (*Francis et al., 2016*) and are upregulated after irradiation (*Akdemir et al., 2007*). X-ray induced expression of *Irbp18* and *Xrp1* depends on p53, a master transcriptional regulator of DDR and apoptosis, which itself is upregulated after exposure to X-rays (*Akdemir et al., 2007*). *Irbp18, Xrp1*, and *p53* were enriched in every scRNA-seq cluster after irradiation, with relatively homogenous expression across them at 4000 rad (HHI scores of 0.031, 0.030, and 0.030, respectively) (*Figure 5A*, right). We confirmed the homogenous expression patterns of *p53* and *Xrp1* using HCR in the wing disc at 0 and 4000 rad (*Figure 5B, C, Q and R*; *Figure 5—figure supplement 1C* for increased gain at 0 rad).

Two TFs with heterogeneous expression after irradiation were the pro-regenerative TF *Ets21C* (*Worley et al., 2022*) and *Dif*, which is activated by the Toll pathway (*Ip et al., 1993*). In scRNA-seq, *Dif* was expressed most highly in the pouch of the disc, though it was also present in other regions in the 4000 rad condition (HHI = 0.051) (*Figure 5A* right, 5T left). An HCR of *Dif* confirmed this pattern of expression (*Figure 5T* right). *Dl*, the other TF downstream of the Toll pathway was present in all scRNA-seq clusters at 4000 rad (*Figure 5—figure supplement 1A*). In the 4000 rad condition, *Ets21C* scRNA-seq expression was concentrated in the hinge, with lesser but present expression in other regions of the disc (HHI = 0.046) (*Figure 5A* right, 5 H left). An HCR of *Ets21C* in irradiated wing discs confirmed this, with regions of high expression present in both the dorsal and ventral hinge (*Figure 5H* right). We next sought to determine if *Ets21C* required *p53* for its induction after X-ray exposure and performed HCR on *Ets21C* in *p53* mutant wing discs. *Ets21C* showed little to no expression before or after X-ray exposure in *p53* mutant wing discs (*Figure 5J and K*), indicating its dependence on *p53* for X-ray induction. This is consistent with previous work showing that p53 can activate the JNK pathway (*Shlevkov and Morata, 2012*), and that *Ets21C* expression is JNK dependent, at least in tissues with overgrowth perturbations (*Toggweiler et al., 2016*), although it is also possible that p53 directly activates Ets21C transcription but only in some regions of the disc.

Another TF that drew our interest was *dysf*, which is required for the development of the larval tracheal system and adult leg joints (*Córdoba and Estella, 2018*; *Jiang and Crews, 2003*) but has yet to be described in the context of exogenous stressors. In scRNA-seq, *dysf* was less concentrated in expression across clusters relative to *Ets21C* and *Dif*, but more so than *p53, Xrp1, or Irbp18* at 4000 rad (*Figure 5A*, right). HCR of *dysf* revealed expression in the dorsal and anterior-ventral hinge of the disc at 0 rad (*Figure 5L*, left), which was reflected in the scRNA-seq expression pattern (*Figure 5L*, right). At 4000 rad, *dysf* was additionally induced in all major PD regions (HHI = 0.037) (*Figure 5M*, right, see *Figure 5M*, left, for matched scRNA-seq expression). However, this induction was only observed in a subset of discs, as some irradiated discs showed no additional *dysf* expression after irradiation (*Figure 5N*). Importantly, irradiated discs with no induced *dysf* expression did show increased *Ets21C* and *p53* expression similar to those with X-ray-induced *dysf* (*Figure 5D and I*), supporting the fact

that they weren't excluded from X-ray exposure. These discs also displayed the developmentally regulated expression pattern of *dysf* present in unirradiated discs, indicating that the HCR probes used in this experiment were capable of detecting *dysf* RNA. The reason for the dramatic difference in *dysf* induction between subsets of discs is not known. From the scRNA-seq datasets, we know that X-ray-induced *dysf* is present in cells from both male and female larvae. One possible explanation is that discs may vary slightly in maturity and that *dysf* might simply not be induced in discs that are slightly more mature, as has been observed for several damage-responsive genes (*Harris et al., 2020*).

Interestingly, in those discs with additional *dysf* expression at 4000 rad, X-ray induced *dysf* RNA was localized to nuclei (*Figure 5M*, *Figure 5—figure supplement 1B* for higher magnification). In contrast, the regions of *dysf* expression present at 0 rad had no discernible biased subcellular localization of the RNA in either condition. The specific nuclear localization of X-ray-induced *dysf* RNA was unique among the genes we visualized with HCR.

When unirradiated *p53* mutant discs were probed for *dysf,* the wild-type expression pattern was present (*Figure 5O*). After X-ray exposure, no *p53* mutant discs were found to have additional *dysf* expression, though the expression pattern found in unirradiated discs was present (*Figure 5P*). Together, these results indicate that the pattern of *dysf* expression found in unperturbed wing discs is *p53* independent and the RNA is not restricted to the nucleus. In contrast, X-ray-induced *dysf* expression depends upon *p53,* its RNA is mostly localized to the nucleus and appears to occur in most but not all wild-type discs.

Taken together, these results indicate that, for IR-responsive transcription factors, some are induced homogeneously while others are expressed in specific regions. Moreover, at least two of the IR-induced transcription factors, *Ets21C* and *dysf,* require p53 for their induction, as has been shown for *Xrp1* by others (*Brodsky et al., 2004*; *Khan and Baker, 2022*). It is therefore likely that p53 functions in combination with other region-specific factors to enable IR-induced expression of *Ets21C* and *dysf.*

## Cell-cycle-based clustering of cells reveals an emergent, high-*tribbles* transcriptional state with increased expression of several genes encoding secreted proteins

So far, we have characterized heterogeneity at the territorial level. However, from prior studies, we know that there must be heterogeneity in cellular responses to radiation even within territories. This is because, at the dose of radiation used, some cells die while the remaining cells survive (*Haynie and Bryant, 1977*) and typically resume proliferation. It is likely that these two classes of cells are interspersed among each other. For clustering thus far, we have used Seurat v5's standard pipeline, which calls upon the 2000 most variable genes in the dataset for clustering and dimensionality reduction, including UMAP. This results in clusters that are stratified based on the greatest sources of transcriptional variation at the global level. In the wing disc, the major sources of variation appear to be in genes that differ in expression along the PD axis, and thus the UMAP bears some similarities to the layout of the wing disc itself, where individual clusters are drawn from particular regions of the disc (*Figure 1D*). In such an approach, cells that differ in response to radiation are likely to be found in each cluster.

We, therefore, considered other methods of clustering that might emphasize differences in transcriptional response that are not dependent upon the location of the cell within the disc. One approach would be to base the clustering on genes that are known to vary between cells in a location-independent manner. Our previous work has shown that in populations of relatively homogeneous cells, such as the myoblasts of the wing disc, cell cycle genes can drive cell clustering (*Everetts et al., 2021*), and IR is known to affect the distribution of cells within the cell cycle.

To explore the relationship between cell-cycle state and X-ray response in our data, we applied a cell-cycle-based clustering approach. We performed dimensionality reduction and clustering on a manually curated list of 175 cell-cycle genes (*Supplementary file 6*). We note that separating cells into distinct phases of the cell cycle based on transcriptional profile alone is a notoriously difficult task for which several approaches have been developed (comprehensively reviewed in *Guo and Chen, 2024*). One type of approach, like the one used here, relies upon manually selected cell cycle marker genes to classify cells into different cell cycle phases. The 175 cell cycle genes used here are drawn from several sources (see methods) and represent our best guess at genes with core functions in different phases of the cell cycle.

For clustering, we applied the Louvain algorithm with a resolution parameter of 0.5, using the first 30 principal components (PCs, cell embeddings transformed with CCA-based integration) of the 175 cell cycle genes as variables, resulting in 6 total clusters. For visualization of clusters, we ran UMAP on these same 30 PCs (*Figure 6A–A'*). Expression of the *tribbles* (*trbl*), a gene that antagonizes the G2/M transition and *PCNA*, which encodes a protein that functions during DNA replication is observed in different regions of the UMAP (*Figure 6B, B', C and C'*). Of the 175 cell cycle genes used for clustering, cluster 3 was high in expression of S-phase genes, including *PCNA, Ts, Claspin,* and *Mcm5* (*Figure 6D* and *Supplementary file 6*). Cluster 5 was also high in many of the same S-phase genes, including *PCNA*, albeit to a lesser extent, but was also marked by three M-phase genes, *aurA, aurB,* and *bora* (*Figure 6D* and *Supplementary file 6*). The high S-phase markers of cluster 3 (hereafter 'high-*PCNA* cluster-a') are consistent with this cluster containing cells in the G1/S phase of the cell cycle. The high but slightly lower levels of these S phase markers, including *PCNA* in cluster 5, along with the expression of *aurA, aurB,* and *bora,* which are important for mitosis (*Giet and Glover, 2001*; *Glover et al., 1995*; *Hutterer et al., 2006*), is consistent with this cluster containing cells in late S/G2 (hereafter 'high-*PCNA* cluster-b'). There was a notable reduction in the proportion of cells belonging to the high-*PCNA* cluster-a from 0 rad (~21% of cells) to 4000 rad (~8% of cells) (*Figure 6E*), indicating a reduction in the proportion of cells in this putative G1/S phase transcriptional state after X-ray exposure.

Cluster 4 was marked by the high expression of *trbl* (*Figure 6D*), which encodes a protein with sequence similarity to kinases, but which lacks kinase activity and generates a G2/M cell cycle arrest by inducing the degradation of the mitosis-promoting Cdc25-orthologous proteins String (Stg) and Twine (Twe) (*Mata et al., 2000*). Cluster 4 (hereafter 'high-*trbl* cluster') had the highest expression of *p53*, high levels of genes involved in both DNA synthesis and repair, such as *RPA1, RPA2, RPA3, and Spn-a,* and relatively low expression of *stg* (see *Supplementary file 6* for complete list of cell-cycle cluster markers). High *trbl* and low *stg* expression are consistent with this cluster containing cells arrested, or entering arrest, at the G2/M phase of the cell cycle. The high-*trbl* cluster contained few cells at 0 rad (~2%) with markedly increased representation at 4000 rad (~18%) (*Figure 6E*). The large increase in the number of cells in this transcriptional state after irradiation is compatible with the large increase in G2/M cells after irradiation observed in vivo by other groups using the cell cycle reporter FUCCI and DNA content quantification using FACS (*Ruiz-Losada et al., 2022*). For cluster stability at different clustering resolutions, see *Figure 6—figure supplement 1*. For cluster relationships, see *Figure 6—figure supplement 2*.

To see how cells belonging to the high-*PCNA* and high-*trbl* clusters were distributed amongst clusters that were previously generated at a global level from the 2000 most variable genes in the dataset, we transferred cell-cycle-based cluster annotations onto our previously clustered data. We found that cells belonging to the high-*PCNA* and high-*trbl* clusters were distributed among each of the seven broad PD regions (*Figure 6F and F'*). As expected, within those broad regions, *PCNA* and *trbl* expression matched the distribution of high-*PCNA* and high-*trbl* cluster identities (*Figure 6G–H'*). Despite high-*PCNA* cells also occupying each major PD region of our single-cell data, EdU signal was notably higher in the hinge of irradiated discs compared to other regions (*Figure 1M*). To confirm the possibility that high-*PCNA* cells may be present across irradiated discs while DNA synthesis occurs primarily in the hinge, we performed HCR against *PCNA* in irradiated discs and found that cells expressing high levels of *PCNA* were present across the PD axis (*Figure 6—figure supplement 3D, G*, quantified in *Figure 6—figure supplement 3I*). We also visualized *bora*, a strong marker for the high-*PCNA*-cluster-b, and found *bora*-expressing cells across the PD axis of irradiated discs (*Figure 6—figure supplement 3E, H*). Similarly, cells that upregulated *trbl* were also found in most regions of the disc as visualized by expression of a *trbl*-GFP reporter (*Figure 6—figure supplement 3A, B*) and by HCR (*Figure 6—figure supplement 3C, F, I*). Taken together, these findings suggest that cells belonging to the high-*trbl*, high-*PCNA* a, and high-*PCNA* b clusters are interspersed throughout the PD axis of irradiated discs.

Next, we focused on differences in genes that were highly induced by X-ray irradiation (HIX; defined as X-ray induced genes >log$_2$FC 1 between conditions, n=359) between cell cycle clusters. At 4000 rad, both high-*PCNA* clusters had relatively low levels of many X-ray induced genes, including those involved in apoptosis (*rpr, hid, Corp, egr*). This was a general trend, with the average scaled expression across HIX genes being slightly lower in the high-*PCNA* clusters than in other clusters

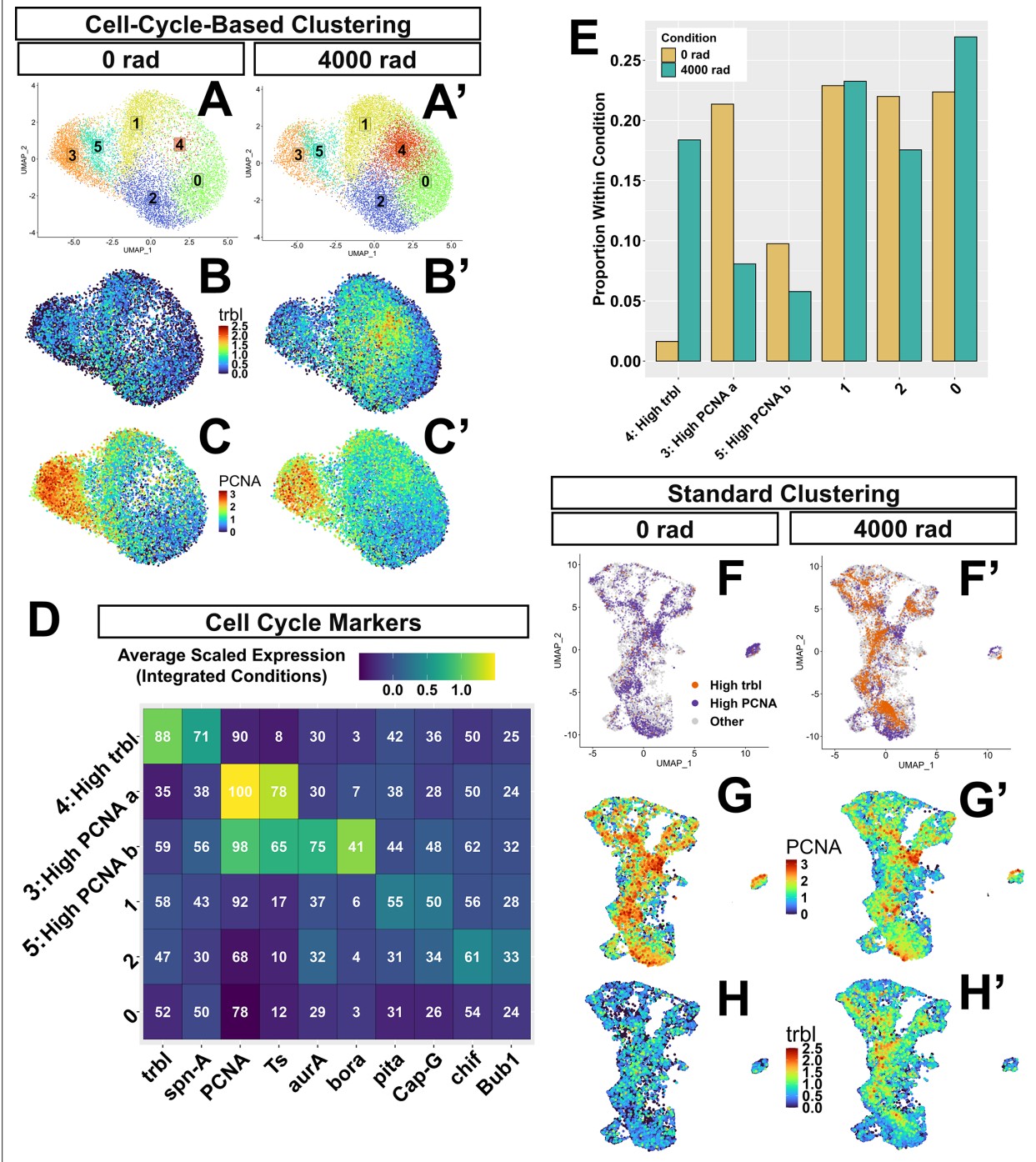

**Figure 6.** Clustering of cells on 175 cell cycle genes. (**A–A'**) Cluster UMAPs of data processed and clustered on cell cycle genes at 0 rad (**A**) and 4000 rad (**A'**). (**B–C'**) Expression of *trbl* and *PCNA* in this UMAP object at 0 rad (**B, C**) and 4000 rad (**B', C'**). (**D**) Heatmap showing average scaled expression of cell cycle marker genes in each cell cycle cluster considering both integrated conditions. Numbers are the percent of cells expressing each gene in each cluster. (**E**) Bar plot showing the proportion of cells each cluster contributes to total cells at 0 rad (cream) and total cells at 4000 rad (teal) conditions. (**F–H'**) High-*trbl* cluster (in orange) and both high-*PCNA* clusters (in purple) from panels A-A' mapped onto the standard UMAP at 0 rad (**F**) and 4000 rad (**F'**). (**G–H'**) Standard UMAPs showing expression of *PCNA* and *trbl* at 0 rad (**G, H**) and 4000 rad (**G', H'**).

The online version of this article includes the following figure supplement(s) for figure 6:

**Figure supplement 1.** Cluster tree showing cluster stability at different clustering resolutions.

**Figure supplement 2.** Dendrogram of cell-cycle-based cluster relationships.

**Figure supplement 3.** Expression of cell-cycle-based cluster markers.

(*Figure 7A*). Both of these clusters had few positive X-ray induced markers (*Supplementary file 7*). In comparison, the high-*trbl* cluster showed a strong enrichment for many HIX genes, having the highest average scaled expression across all of these genes (303/359) compared to other clusters in the 4000 rad condition (*Figure 7A*).

The high-*trbl* cluster was enriched in HIX genes involved in apoptosis (*rpr, hid, Corp, egr*), DNA damage repair (e.g. *rad50, Irbp18, Ku80, mre11*) and ROS-related genes (*GstE6, GstE7, GstD1*). Additionally, this cluster was enriched in HIX genes encoding a subset of TFs (*Ets21C, dysf,* and *Dif),* and secreted proteins *(Pvf2, upd2, upd3,* and *spz)* (for the complete list of markers, see *Supplementary file 7*). Most of these gene categories were represented among the top 24 HIX genes with highest expression in the high-*trbl* Cluster (*Figure 7B*). Though the majority of HIX genes were highest in the high-*trbl* cluster, 56/359 were generally more homogenous across cell-cycle-based clusters (*Figure 7C*), with none having relative expression in other clusters as high as those enriched in the high-*trbl* cluster (*Figure 7A and C*). These results suggest that a high-*trbl* transcriptional cell-cycle state, which accounts for approximately 18% of cells, is disproportionately associated with many X-ray induced changes in gene expression.

The high-*trbl* cluster was also enriched in two noteworthy HIX genes encoding secreted proteins: *Swim* and *Arc1. Swim* is a secreted protein that binds to Wingless and has been proposed to aid in its spreading, though this point is contended (*McGough et al., 2020*; *Mulligan et al., 2012*; *Simões et al., 2022*). *Arc1* is similar to retroviral gag proteins and has been primarily studied as a regulator of neural plasticity in *Drosophila* but was recently shown in a *RasV12* tumor model to be expressed in tumor-associated hemocytes, with a loss of function being associated with larger tumors, decreased pJNK, and decreased Dcp-1 activity in tumors (*Khalili et al., 2023*).

To determine if increased *trbl* expression affected cell survival after X-ray irradiation and whether it could alter expression of HIX genes enriched in the high-*trbl* cluster, we induced *trbl* for 24 hr prior to irradiation and during recovery using a temperature-sensitive gal4-based system (*McGuire et al., 2003*; *Figure 7—figure supplement 1A*). We observed an obvious reduction in cell death in irradiated discs overexpressing *trbl* compared to controls expressing GFP (*Figure 7—figure supplement 1B–F*). The reduction in cell death observed in *trbl* over-expressing discs is consistent with prior observations that X-ray-induced cell death in the wing disc is reduced when genes that promote G2-M transition are knocked down and cells are arrested in G2 (*Ruiz-Losada et al., 2022*). We also examined effects on two HIX genes with high enrichment in the high-*trbl* cluster: *Swim* and *CG15784*. For both genes, we found that overexpression of *trbl* in the pouch resulted in little or no difference in their expression before or after irradiation compared to controls (*Figure 7—figure supplement 2A–R*).

In conclusion, our results are consistent with transcriptional states that likely correspond to early and late S-phase being reduced in proportion after X-ray exposure, and cells accumulating in what might be a normally uncommon transcriptional state that could correspond to a G2/M arrest. Cells in the putative G1/S and S/G2 states have relatively low expression of many HIX genes. In contrast, cells in the putative G2/M stalled state show relatively high expression of many HIX genes and seem to be responsible for a disproportionate amount[1] of X-ray-induced gene expression. Increased expression of *trbl* does seem to offer some radioprotection, but in itself, does not result in increased expression of two of the HIX genes that we examined.

## Simultaneously visualizing two levels of heterogeneity

To compare the level of heterogeneity that was observed using the two clustering approaches for individual genes, we plotted their two HHI scores on a two-dimensional plot where the X-axis shows the score obtained using cell-cycle-based clustering and the Y-axis shows the score obtained from the clusters obtained with the standard Seurat pipeline (*Figure 8*). For these plots, we used transformed HHI scores where the original HHI scores, which ranged from 1 /N to 1 (where N represents the number of clusters), were re-mapped to a range from 0 to 1. Importantly, the two parameters cannot be truly orthogonal since the cell cycle state likely makes some contribution to the clustering using the standard pipeline and because a cell's location in the disc might affect its behavior with respect to the cell cycle. Additionally, the distribution of HHI scores with respect to any parameter changes with the number of clusters being analyzed - thus, the values on the two axes are not directly comparable. With these caveats in mind, these plots provide us with a sense of the extent of heterogeneity of expression that each gene displays when assessed in each

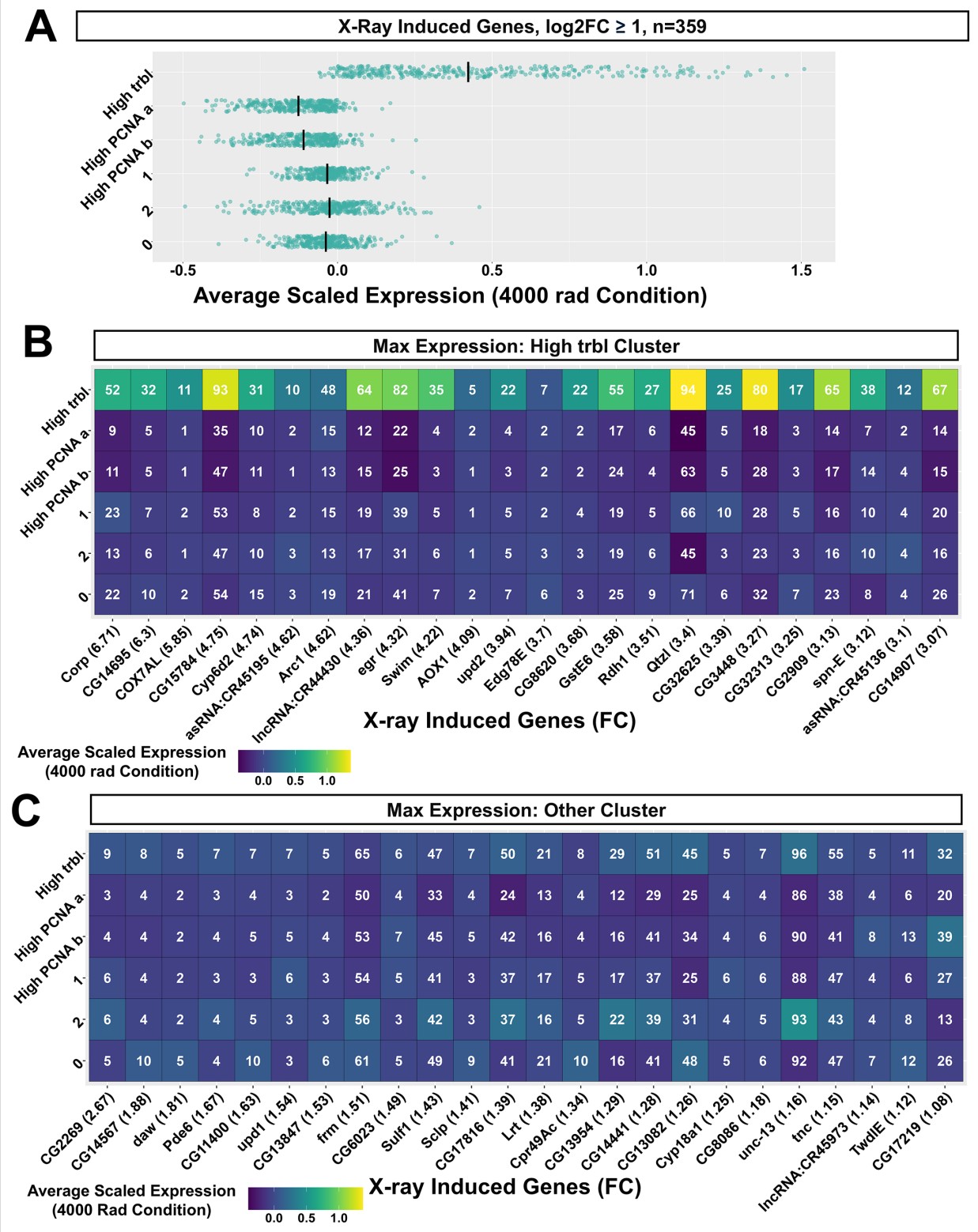

**Figure 7.** Differences in X-ray induced gene expression between cell-cycle-based clusters. (**A**) Mean scaled expression of X-ray induced genes with avg log$_2$FC ≥1 between conditions. Mean-scaled expression values were calculated from cell-cycle-based clusters using cells from the 4000 rad condition only. Each row contains the same genes. (**B**) Heatmap showing the average scaled expression of the top 24 X-ray-induced genes with highest expression in the High-*trbl* Cluster. (**C**) 24 of the genes with maximum expression in a cell-cycle-based cluster other than the High-*trbl* Cluster. Numbers are the percentage of cells expressing the gene in each cluster. Only genes with ≥5% expression in any cluster at 4000 rad were selected from the initial 359.

*Figure 7 continued on next page*

*Figure 7 continued*

The online version of this article includes the following figure supplement(s) for figure 7:

**Figure supplement 1.** Effects of *trbl* overexpression on cell death.

**Figure supplement 2.** Effects of *trbl* overexpression on HIX genes *Swim* and *CG15784*.

of these two ways and allows us to compare the relative HHI scores of genes on each axis to each other.

As can be seen from the plots, genes encoding components of the DDR and regulators of ROS levels, apoptosis, and cell cycle progression have low HHI scores with little variation when standard

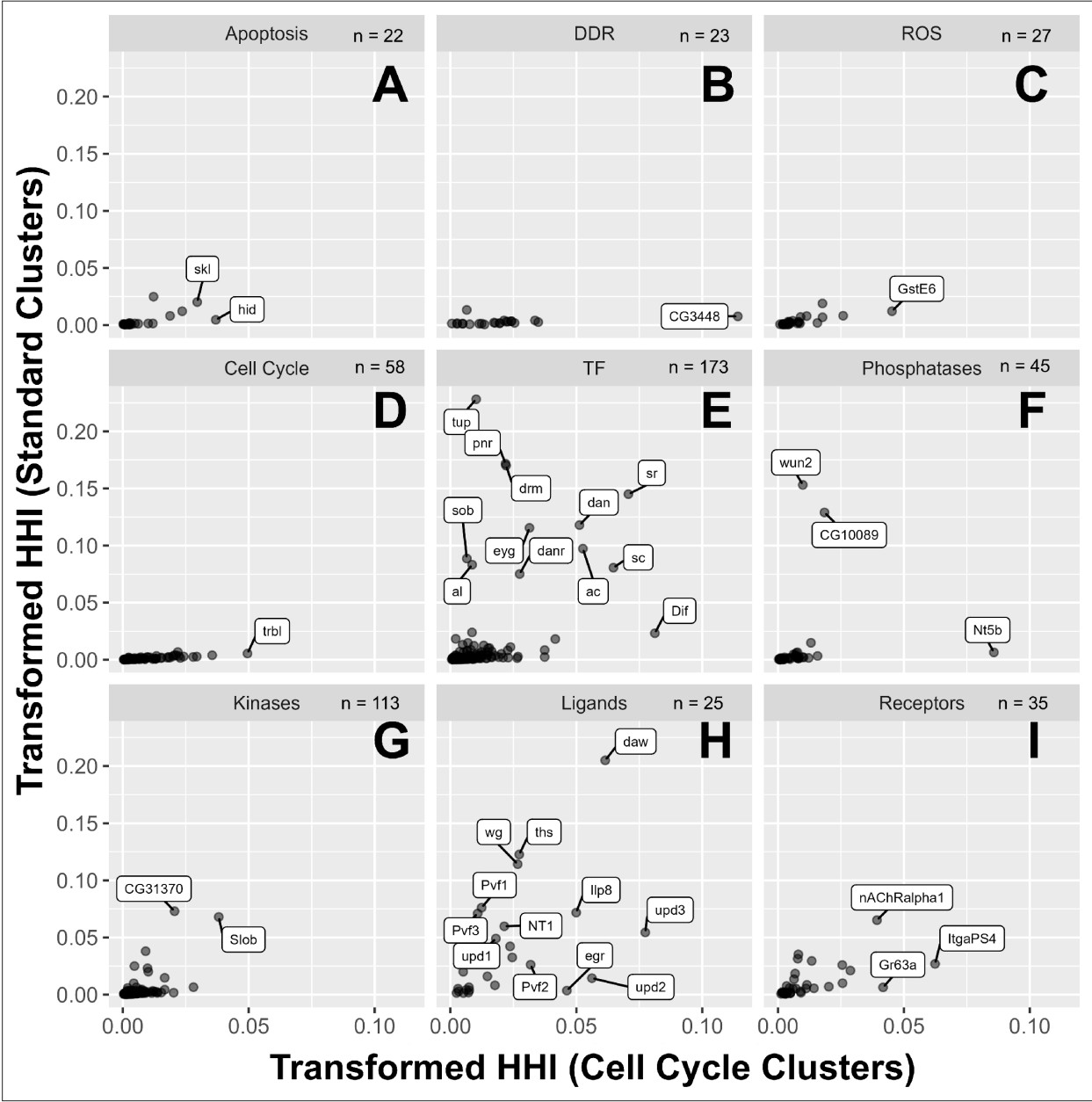

**Figure 8.** Herfindahl-Hirschman Index (HH)I scoring of categorical genes in standard clusters vs cell-cycle-based clusters. (**A–I**) Transformed HHI scores calculated using the 35 standard clusters (y-axis) and the six cell-cycle-based clusters (x-axis). Genes used were the 521 categorical genes shown previously in *Figure 3A*. Scores were calculated using cells from the 4000 rad condition only. Genes displayed encode for proteins involved in apoptosis (**A**), DNA damage response (DDR) (**B**), reactive oxygen species (ROS) (**C**), cell cycle (**D**), transcription factors (TFs) (**E**), phosphatases (**F**), kinases (**G**), ligands (**H**), and receptors (**I**). Genes are labeled where space permits but were otherwise chosen for labeling arbitrarily.

clustering is used (*Figure 8A–D*). These genes are likely to be expressed homogenously across regions in the disc. In comparison, when HHI scores are generated for genes in these categories using cell-cycle-based clustering, their HHI scores have greater variation, with some genes having much higher scores than others (e.g. *CG3448* in DDR). Such genes should, in theory, display little heterogeneity in their expression across different regions of the disc while showing greater variation in expression associated with cell-cycle transcriptome state. Genes encoding TFs (*Figure 8E*) and ligands (*Figure 8H*) contain many genes with relatively high scores on either or both axes, indicating these genes are most heterogeneous with regard to regions (e.g. *tup*), cell cycle transcriptome state (e.g. *Dif*), or a combination of both (e.g. *sr*). Phosphatases, kinases, and receptors had an intermediate range of HHI scores with regard to both axes (*Figure 8F, G and I*).

## Discussion

An important shift in our understanding of biological systems in recent decades has been an increasing awareness of heterogeneity. This applies at multiple levels – from molecules to cells. At the level of cells, even individual cells in microbial communities have considerable phenotypic diversity, especially in response to changes in environmental conditions (*Ackermann, 2015*). Thus, it is likely that cells in mammalian tissues, which are often composed of many different types of cells, will display diverse responses to external stressors, such as ionizing radiation. Only recently have single-cell approaches

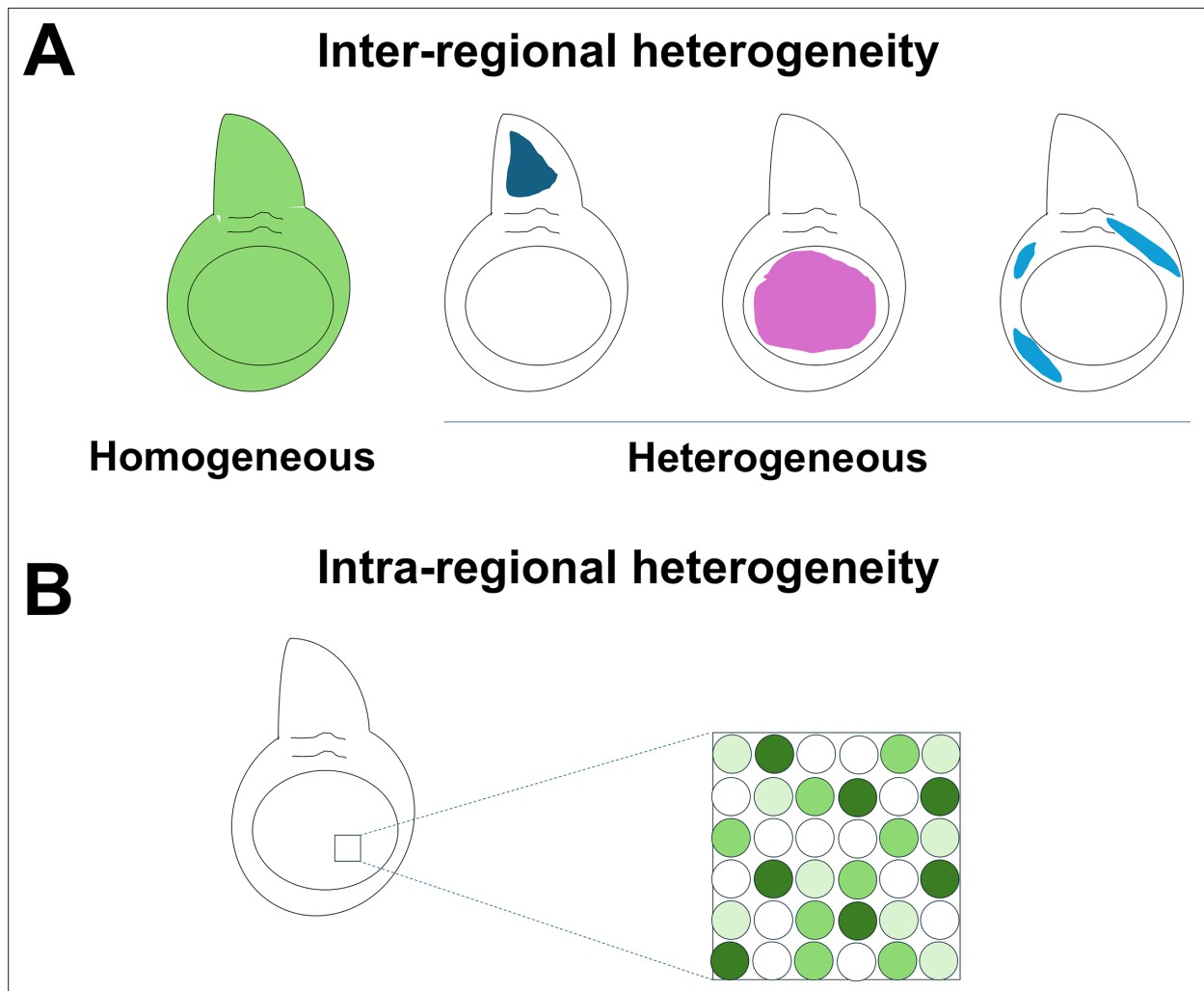

**Figure 9.** Two types of heterogeneity following X-ray irradiation. (**A**) Inter-regional heterogeneity. Many genes are expressed uniformly while some are expressed in specific territories following irradiation. (**B**) Intra-regional heterogeneity. In the example shown, a gene is expressed at high levels in some cells, intermediate levels in some, and no expression in others. Cells with different levels of expression are interspersed.

been used to study the effects of ionizing radiation on complex tissues, for example (*Lu et al., 2023*; *Mills et al., 2025*; *Morral et al., 2024*; *Mukherjee et al., 2021*; *Sheng et al., 2020*; *Yuan et al., 2023*). These studies have mostly focused on changes in cellular composition following radiation, or on characterizing particular subpopulations of cells, rather than attempting to address the overall types of heterogeneity in IR response within and between cell types.

To make the study of heterogeneity of cellular responses to radiation more tractable, we have used a relatively simple and homogenous tissue, the *Drosophila* wing disc which is mostly composed of two different cell types (reviewed by *Tripathi and Irvine, 2022*). Each wing disc is composed of a little over 30,000 epithelial cells (*Martín et al., 2009*) and approximately 2500 myoblasts (*Gunage et al., 2014*). This has allowed us to look for differences in the way cells in different parts of the wing disc respond to radiation without the complication of diverse cell types. Our studies have uncovered two kinds of heterogeneity: inter-regional and intra-regional (*Figure 9*). Inter-regional heterogeneity is where some genes are upregulated far more in some territories of the wing disc than in others (*Figure 9A*). Intra-regional heterogeneity is manifest in all major PD regions of the wing disc. Here, cells with different types of radiation-responsive transcriptomes are interspersed among each other (*Figure 9B*).

## Using the Herfindahl-Hirschman Index to assess heterogeneity in single-cell datasets

Studies that use scRNA-seq or multiomics frequently characterize differences in gene expression between clusters but typically do not use quantitative approaches to examine the extent of heterogeneity between clusters on a genome-wide level. To rank genes with respect to heterogeneity of expression between clusters, we have used the Herfindahl-Hirschman Index (HHI), which is a simple mathematical tool most typically used by economists to study market concentration (*Herfindahl, 1950*; *Hirschman, 1945*). This and other mathematical ways of calculating heterogeneity are summarized by *Steele et al., 2022*. In the context of our application, a low HHI score indicates a lower level of concentration or more homogenous expression, while a higher score indicates that most of the expression occurs in one or a few clusters. While the score of the index for any given gene is not particularly useful in isolation, ranking genes with the HHI has allowed us to identify those genes that are expressed more homogeneously among clusters as opposed to those that are expressed primarily in a few clusters. When we have used the conventional Seurat pipeline, a higher HHI score typically identifies genes that are expressed in a subset of territories in the wing disc, i.e., inter-regional heterogeneity. When applied to clusters generated using a curated set of cell cycle genes, a high HHI score indicates that a given gene is preferentially expressed in a subset of cells with a particular transcriptome state of cell cycle genes. This type of intra-regional heterogeneity is observed in all PD regions of the disc. Thus, these two types of heterogeneity can be shown simultaneously using a two-dimensional plot (*Figure 8*). Given that the same tissue can be analyzed using a variety of cellular parameters (e.g. chromatin accessibility, proteomic approaches), cells can be clustered using each of these approaches, and an HHI score for each gene could be calculated with respect to that parameter. In principle, each gene could then be assigned a multi-dimensional HHI score that reflects its heterogeneity with respect to each parameter.

## Characterization of inter-regional heterogeneity

Since DNA damage, as assessed by p-H2Av immunofluorescence, is distributed throughout the disc, it is unsurprising that most genes upregulated in response to DNA damage are also expressed in most regions of the disc. These include genes encoding proteins that repair DNA damage, inactivate ROS, as well as those that regulate cell cycle responses and apoptosis in response to DNA damage. This also applies to the key transcription factor considered to regulate many of the transcriptional responses to DNA damage, p53. We find that heterogeneous expression with respect to geographical territories in the wing disc is most evident among genes encoding transcription factors and secreted ligands and is also found occasionally among receptors and signal transducers, such as kinases and phosphatases.

Why are these genes expressed heterogeneously? By manually inspecting gene expression, we find that in many cases, the expression pattern following irradiation is similar to that in unirradiated discs. Like our approach using the HHI, we have used a Euclidean distance measure in seven-dimensional space to rank the difference in expression pattern with respect to the seven major regions of the disc

under the two conditions (unirradiated versus irradiated) (*Figure 3C*). Genes with an especially low Euclidean distance have more similar patterns of expression between the two conditions. Clear examples of this are *dawdle* (*daw*) which encodes a ligand for an activin receptor, and *upd1* which encodes a ligand upstream of the JAK/STAT pathway. Both genes have similar patterns of expression under the two conditions but are expressed at higher levels in the same regions after irradiation. Why might this happen? One possibility is that the increased expression of these genes requires both a region-specific transcription factor and a transcription factor induced by irradiation. An alternative possibility is that the chromatin state of such genes might reflect their pattern of expression in the unirradiated disc; a radiation-induced transcription factor might bind more readily to genes that already have more accessible chromatin. Both explanations are consistent with our observation that the non-uniform upregulation of several radiation-induced genes, including *upd2*, *upd3,* and *Ets21C* requires the function of *p53*, which is induced by X-ray irradiation yet expressed in all territories of the disc.

For genes, such as the JAK/STAT ligands *upd2* and *upd3*, expression is clearly non-uniform after radiation. Yet in the unirradiated condition, expression is minimally present or not detected. This pattern of induction is more difficult to explain. It is possible that *upd2* and *upd3* might be expressed at extremely low levels in unirradiated discs in patterns similar to their pattern of induction. Also possible is that the chromatin state of these genes, even if they are not expressed in the unirradiated disc, shows inter-regional heterogeneity. This non-uniform induction was also observed with the transcription factor *Ets21C*, which is known to be necessary for regeneration of the wing pouch after its ablation (*Worley et al., 2022*). Our identification of many genes that likely play an important role in IR response with inter-regional differences in their induction indicates that this phenomenon is relatively widespread and that its mechanistic basis merits further investigation.

## Intra-regional heterogeneity: cells with a particular cell cycle transcriptome have the highest levels of expression of damage-responsive genes after irradiation

Since the standard Seurat pipeline, when applied to the wing disc, generated clusters that most reflect a cell's location in the wing disc, we tried alternative approaches that could separate cells into clusters that were agnostic to their location in the disc. Using a curated set of cell cycle genes, we were able to separate cells into six clusters. One cluster accounted for only 2% of cells in the unirradiated state but 18% of cells following irradiation. This cluster was marked by high expression of the *trbl* gene which encodes a pseudokinase that promotes the degradation of the Cdc25 ortholog String (*Mata et al., 2000*). Based on their transcriptome, cells in this cluster are likely to be in G2 and prevented from entering mitosis.

Unexpectedly, *p53*, as well as genes involved in DNA repair, apoptosis, and inactivation of ROS were all most highly expressed in this cluster at 4000 rad, as were the majority of radiation-induced genes. Thus, rather than being expressed uniformly in all cells, expression of many radiation-induced genes shows strong intra-regional heterogeneity with the highest expression found in a subpopulation that occupies a particular cell cycle transcriptome state. This same high-*trbl* cluster also expresses higher levels of the transcription factor *Ets21C* as well as genes encoding a variety of secreted ligands, such as *Pvf2*, *upd2*, *upd3*, and *spz*. Studies of regeneration in recent years have identified situations where subsets of cells function to organize regenerative proliferation by secreting factors that promote the survival and proliferation of other cells (*Aztekin et al., 2019*). Indeed, following ablation of the wing pouch, a localized subpopulation of cells likely arrested in G2 (*Cosolo et al., 2019*) promotes the proliferation of surrounding cells (*Worley et al., 2022*). In the case of diffuse damage, such as that elicited by X-ray exposure, the analogous population of organizing cells could be dispersed throughout the disc and interspersed with other cells. Cells belonging to the high-*trbl* cluster could potentially represent such a population.

We have also shown that increasing *trbl* expression reduces the extent of apoptosis suggesting that the cells in the high-*trbl* cluster could also be more radioresistant. This is consistent with previous work that has shown that cells that seem stalled in G2 after exposure to the TNF ortholog *egr* are relatively resistant to JNK-induced apoptosis (*Cosolo et al., 2019*) and that reducing activity of *string*, which promotes G2/M progression, confers some level of protection from X-ray-induced apoptosis (*Ruiz-Losada et al., 2022*). We also found that increasing *trbl* expression did not increase the expression

of two other HIX genes implying that other factors, that may or may not be cell-cycle dependent, are necessary for their upregulation.

### Functional consequences of heterogeneity

Many of the genes that display inter-regional heterogeneity have been implicated previously in regulating cell survival and proliferation, as well as in activating mechanisms that function during regeneration. These include the *upd2* and *upd3* genes, and the transcription factor *Ets21C*. Previous work has shown that a region of the wing disc, the dorsal hinge, is more resistant to apoptosis following irradiation, and that this resistance is dependent upon increased JAK/STAT signaling in that region (**Verghese and Su, 2016**). It is, therefore, likely that many of the examples that we have observed of inter-regional heterogeneity have functional consequences. A systematic disruption of all genes that display heterogeneous patterns of expression would give us a more complete view of this phenomenon.

The observation that the majority of HIX genes are most expressed in a subset of cells defined by the expression of particular cell cycle genes was an unexpected finding of our analysis. This could imply that cells must be in this state to most effectively turn on these genes. Alternatively, the suite of genes that are expressed most after irradiation could push the cells to a particular cell cycle state. Such a state might be more conducive to the repair of cellular damage and to promote local and systemic responses to radiation that facilitate recovery. Further study of this phenomenon could enable ways of using cell cycle manipulations to alter radiosensitivity.

### Concluding remarks

Our studies have focused on one time point at one developmental stage in a relatively simple tissue. The levels of heterogeneity that we have uncovered here suggest that heterogeneity in response to radiation must be far more widespread and complex in tissues, such as mammalian organs that are composed of many different types of cells. The approach and the methods of analysis used here should be applicable, with appropriate modifications, to these more complex situations.

## Materials and methods

**Key resources table**

| Reagent type (species) or resource | Designation | Source or reference | Identifiers | Additional information |
|---|---|---|---|---|
| Strain, strain background (*Drosophila melanogaster*) | Oregon R | Bloomington *Drosophila* Stock Center | Stock #25211; RRID:BDSC_25211 | |
| Strain, strain background (*Drosophila melanogaster*) | p53[5 A-1-4] | Bloomington *Drosophila* Stock Center | Stock #6815; RRID:BDSC_6815 | |
| Strain, strain background (*Drosophila melanogaster*) | UAS-trbl | Bloomington *Drosophila* Stock Center | Stock #58493; RRID:BDSC_58493 | |
| Strain, strain background (*Drosophila melanogaster*) | UAS-GFP | Bloomington *Drosophila* Stock Center | Stock #4776; RRID:BDSC_4776 | |
| Strain, strain background (*Drosophila melanogaster*) | trbl-GFP | Bloomington *Drosophila* Stock Center | Stock #61654; RRID:BDSC_61654 | |
| Antibody | Rabbit anti-H2AvD-pS13 (polyclonal) | Rockland | Cat #600-401-914; RRID:AB_828383 | (Dilution used = 1:250) |
| Antibody | Rabbit anti-Dcp-1 (polyclonal) | Cell Signaling | Cat #9578; RRID:AB_2721060 | (Dilution used = 1:250) |
| Antibody | Rabbit anti-PHH3 (polyclonal) | Millipore | Cat #06–570; RRID:AB_310177 | (Dilution used = 1:500) |
| Antibody | Rat anti-Zfh2 (polyclonal) | Chris Doe; **Tran et al., 2010** | https://doi.org/10.1242/dev.048678 | (Dilution used = 1:200) |
| Antibody | Rabbit anti-GFP (polyclonal) | Torrey Pines Biolabs, Inc | Cat #TP401 | (Dilution used = 1:500) |
| Antibody | Goat anti-rat Alexa Fluor 555 (polyclonal) | Thermo Fisher Scientific | Cat #A-21434; RRID:AB_2535855 | (Dilution used = 1:500) |

*Continued on next page*

*Continued*

| Reagent type (species) or resource | Designation | Source or reference | Identifiers | Additional information |
|---|---|---|---|---|
| Antibody | Goat anti-rabbit Alexa Fluor 555 (polyclonal) | Thermo Fisher Scientific | Cat #A-21428; RRID:AB_2535849 | (Dilution used = 1:500) |
| Commercial assay, kit | HCR RNA-Fish (v3) (buffers, hairpins, probes) | Molecular Instruments | | |
| Commercial assay, kit | Click-iT EdU Cell Proliferation Kit, Alexa Fluor 647 | Thermo Fisher Scientific | Cat #C10340 | |
| Software, algorithm | Image J / Fiji | https://fiji.sc/ | | |
| Software, algorithm | Kallisto-Bustools (v0.28.2) | Pachter Lab | | https://www.kallistobus.tools/ |
| Software, algorithm | R | R Project for Statistical Computing | | http://www.r-project.org/ |
| Software, algorithm | org.Dm.eg.db (v3.18.0) | Bioconductor | | https://www.bioconductor.org/packages/release/data/annotation/html/org.Dm.eg.db.html |
| Software, algorithm | Seurat (v5) | Comprehensive R Archive Network | | https://cran.r-project.org/web/packages/Seurat/index.html |
| Software, algorithm | clustree | Comprehensive R Archive Network | | https://cran.r-project.org/web/packages/clustree/index.html |

## Fly strains used

Oregon R; p53[5 A-1-4] (BDSC #6815); UAS-trbl (BDSC #58493); UAS-GFP (BDSC #4776); For trbl protein visualization, trbl-GFP (BDSC #61654) was used and signal boosted with anti-GFP.

## Antibodies used

### Primaries

Rabbit anti-H2AvD-pS13 (#600-401-914, Rockland), 1:250; Rabbit anti-Dcp-1 (#9578 S, Cell Signaling), 1:250; Rabbit anti-PHH3 (#06–570, Millipore), 1:500; Rat anti-Zfh2 (Chris Doe; *Tran et al., 2010*), 1:200; Rabbit anti-GFP (TP401 Torrey Pines Biolabs, Inc), 1:500.

### Secondaries

Goat anti-rat Alexa Fluor 555 (A21434, Invitrogen), 1:500; Goat anti-rabbit Alexa Fluor 555 (A21428, Invitrogen), 1:500.

## Single-cell sample collection

### Egg collections and rearing

All single-cell experiments were performed using the 'Oregon R' wild-type fly strain. Egg lays and rearing were performed at 25 °C. Groups of 15 males and 35–40 females were sorted into food vials and allowed to recover from CO2 incapacitation. Flies were then tapped into egg lay bottles containing a grape agar plate topped with a dollop of yeast paste and pre-incubated for 1–2 days to increase egg yield. On the day of egg collection, flies were tapped into egg-lay bottles containing fresh grape agar plates and yeast paste and incubated for 4 hr. After the egg lay period, grape plates were collected, and yeast paste was removed. Eggs were incubated for 24 hr, after which 1st instar larvae were picked using a poker tool dipped in yeast paste for adhesion. 50 larvae were deposited into each petri dish food plate (preparation of food plates described below) and incubated until their time of irradiation/dissection. Larvae were irradiated with 4000 rad or left outside of the irradiator for control. At the end of irradiation, larvae were allowed to recover at 25 °C.

### Preparation of petri dish food plates for rearing and irradiation

For scRNA-seq experiments: Fly food prepared using the Bloomington *Drosophila* Stock Center (BDSC) formula was melted in the microwave and poured to a height of 10 + –2 mm into clear 15 mm

× 60 mm petri dishes. After cooling, condensation was wiped from the lid, and ~1/8th tsp of dry active yeast was added on top of the food. Plates were stored at room temperature for 1–2 days before use.

## Dissections, tissue dissociations, and *FACS*

All microcentrifugations between washes were performed at 5000 RPM (~2000 g) at room temperature. All samples and media were stored on ice unless otherwise noted. Larvae were collected into 1 X PBS and dissected in supplemented Schneider's media (SSM) at room temperature over 1 hr. Wing discs were collected into SSM on ice, pooled into 1.5 ml Eppendorf tubes, and washed with Rinaldini solution. Wing discs were dissociated at 37°C in 0.25% Trypsin EDTA (Sigma T4049) for 10 min. Dissociated cells were washed once in PBS-10% FBS and twice in PBS-1% FBS. Cells were pooled together in a final suspension of 500 µl PBS-1% FBS and passed through a 35 µM filter into a FACS tube. Cell suspensions were sorted on a BD FACSAria Fusion. Forward scatter and side scatter were used to sort out debris and doublets. Propidium iodide (PI) was added to the sample and a Texas Red filter was used to remove low quality/dead cells: PI fluorescence intensity revealed a bimodal distribution of cells, and a gate was drawn between the two distributions. The upper distribution was sorted out. Cell concentrations were determined using a hemocytometer, and brought to an appropriate final concentration in accordance with 10 X Genomics v3.1 guidelines. cDNA libraries were generated using 10 X Genomics v3.1 chemistry and hardware.

## Single-cell data processing

### Library preparation, sequencing, and alignment

All libraries were sequenced at a depth of around 1 billion paired-end reads. Sequences were aligned to *Drosophila* transcriptome v6.55 using Kallisto-Bustools v0.28.2 (Kallisto v0.50.1, Bustools v0.43.2) with default settings. FlyBase gene IDs were translated to gene symbols using org.Dm.eg.db v3.18.0 R package. A table containing all gene symbols used in this dataset, their FlyBase IDs, and their annotation symbols are included as a supplement. ***Supplementary file 8*** contains all annotated genes found in the org.Dm.eg.db v3.18.0 R package, their FlyBase IDs, FlyBase CGs, and their current symbols at the time of analysis, used to translate FlyBase gene IDs to symbols.

### Pre-integration filtering

After alignment, all four datasets (two irradiated, two unirradiated) were individually filtered in the same manner before integrated analysis as follows: (1) Genes which were captured in 3 or less cells, and cells which contained less than 200 unique genes were removed. (2) Unique genes captured per cell were visualized with a density plot (typically used for continuous data, but used here heuristically), revealing roughly bimodal distributions in each sample. One maximum was near zero (representing 'low quality' cells) and the other maxima was between 2000 and 5000 unique genes. A cutoff was drawn at the lowest density value between the two maxima, and the lower half of cells were removed. (3) Cells which were in the 98.5th percentile of unique features and above (potential doublets) were removed.

### Standard integration and clustering of processed datasets

All prefiltered datasets were then integrated using the Seurat v5 (https://satijalab.org/seurat/) integration pipeline: (1) All datasets were merged and normalized (2) The top 2000 most variable genes were scaled (3) PCA was performed on the scaled expression of these 2000 variable genes (4) Cell PC embeddings were transformed with CCA-based integration (5) FindNeighbors() was run using the first 30 transformed PC embeddings (6) FindClusters() was run (default Louvain algorithm) with a resolution of 2, resulting in 35 clusters (7) UMAPs were also generated using the first 30 transformed PCs using runUMAP()-Unless specifically stated, default parameters were used for Seurat processing.

### Cell-cycle-based integration and clustering of processed datasets

All prefiltered datasets were run through the same Seurat v5 integration pipeline as above, with some differences: (1) All datasets were merged and normalized (2) All genes were scaled (3) PCA was performed on the scaled expression of 175 manually curated cell-cycle genes (4) Cell PC embeddings were transformed with CCA-based integration, with the 175 cell-cycle genes used for integration (5)

FindNeighbors() was run on the first 30 transformed PCs (6) FindClusters() was run (default Louvain algorithm) with a resolution of 0.5, resulting in 6 clusters (7) UMAPs were also generated using the first 30 transformed PCs using runUMAP().

The 175 cell-cycle genes were primarily drawn from The Interactive Fly (https://www.sdbonline.org/sites/fly/aignfam/cellcycl.htm) and from a list of fly orthologs of cell-cycle genes used in *Tirosh et al., 2016*, compiled by the Harvard Chan School Bioinformatics Core (here). Some pleiotropic genes, especially those with well-described roles in stress response or cell death (e.g. *buffy*), were excluded from the final list.

Note: Our list of cell-cycle genes originally included *anachromism* (*ana*), but this gene was removed due to its known pattern of restricted regional expression in the wing disc (*Butler et al., 2003*), which drove clustering.

## Selection and plotting of categorical gene lists

Genes belonging to ligands, TFs, receptors, phosphatases, and kinases were drawn directly from FlyBase gene groups (*receptor ligands*, *transcription factors*, *transmembrane receptors*, *phosphatases*, and *kinases*, respectively). Cell-cycle genes were derived as described above. ROS, DDR, and apoptosis genes were derived from a combination of FlyBase gene groups/manual curation of the literature. Genes from these lists that were not detected in our data were filtered out. This filtered list is available as *Supplementary file 3*. Note that 81 genes from *Supplementary file 4* were found to be in more than one categorical list. *Supplementary file 4* contains these 81 genes, the categorical lists they belonged to, and the category they were chosen to be in for analysis. To subset genes in this list induced by X-rays (n=521), we kept all genes expressed in at least 1% of cells in either condition that had a positive log2FC when comparing all cells of the 4000 rad condition to the 0 rad condition, with an adjusted *p*-value <0.05 using Seurat's FindMarkers(). FindMarkers was then run on each gene on a cluster basis, and genes that were present in greater than 5% of cells in any cluster and had a log2FC >0.25 were kept, resulting in 521 genes. This list of genes, with their HHI scores at 0 and 4000 rad, are available as *Supplementary file 5*.

## Differential gene expression analysis

For all differential gene expression analyses, the Wilcoxon Rank Sum test was run via the FindMarkers() function in Seurat v5. When considering significance, the Bonferroni-adjusted p-value was used (adjusted on all genes in the dataset). For *Figures 4A and 5A* p-values were additionally adjusted for the number of clusters tested.

## HHI score calculations

HHI scores for genes were calculated by taking the non-log expression values of clusters (using Seurat's AverageExpression function), transforming expression values into proportions of the sum of all average cluster expressions, and squaring and summing them. For transformation of scores to 0–1, the following formula was used: (Score-(1 /n))/(1-(1 /n)), where n is the number of clusters.

## Other packages used for analysis

The clustree R package was used to generate the cluster stability plot in *Figure 6—figure supplement 1*.

## Timed trbl overexpression

Virgin female flies of the genotype w-;;rn-Gal4, tub-Gal80ts/TM6B, tub-Gal80 were crossed to male w-;; UAS-trbl (BDSC #58493), or w-;; UAS-GFP (BDSC #4776) for controls and allowed to mate at room temperature for a 3–7 days. Eggs were collected at room temperature for 16 hr overnight. After egg lay, larvae were grown at 18° C for 7 days, irradiated with 4000 rads, and allowed to recover at 18° C. For all steps, flies were reared in Bloomington formula food supplemented with yeast paste. For dissection, non-tubby wandering larvae were selected.

## Immunofluorescence

All larvae were reared at 25°C. For each experiment, wandering L3 larvae were picked from non-synchronized stocks and placed into separate vials containing Bloomington formula food. They were

then irradiated with 4000 rad or left outside of the irradiator for control. At the end of irradiation, larvae were allowed to recover at 25 °C. After recovery, 9–15 larvae were picked, dissected in 1 X PBS, and fixed. After fixing for 20 min in 4% formaldehyde in PBS, carcasses then washed three times in 0.1% PBST for 10 min. Carcasses were then blocked in 10% NGS in 0.1% PBST. After blocking, samples were incubated with primary antibodies diluted in blocking solution overnight at 4 °C. The next day, samples were washed three times in PBST for 15 min and then incubated with secondary antibodies diluted in blocking solution at 1:500 for 2.5 hr at 25 °C or overnight at 4 °C. Samples were washed once in 0.1% PBST for 5 min and then incubated with 1:1000 DAPI in PBST for 20 min. Afterwards, samples were washed twice in 1 X PBS for 10 min and wing discs were mounted. Wing discs were imaged on an epifluorescent microscope with an apotome attachment.

## EdU assays

All larvae were reared at 25°C. For each experiment, wandering L3 larvae were picked from non-synchronized stocks and placed into separate vials containing Bloomington formula food. They were then irradiated with 4000 rad or left outside of the irradiator for control. At the end of irradiation, larvae were allowed to recover at 25 °C. After recovery, 9–15 larvae were picked and dissected in Schneider's *Drosophila* Medium (Thermo Fisher 21720024) at room temperature. Discs were incubated in EdU for 30 min. After incubation, discs were fixed in 4% PFA for 15 min. The Click-iT EdU Cell Proliferation Kit, Alexa Fluor 647 (Thermo Fisher C10340) reagents were used and protocol followed.

## In situ hybridization chain reaction

All larvae were reared at 25 °C. For each experiment, about 50 third instar larvae for each condition were placed into vials containing Bloomington formula food for irradiation. Half of the vials were then irradiated at a dose of 4000 rad while the other half were left outside of the irradiator. Larvae were given a 4 hr recovery time at 25 °C after irradiation before the beginning of the dissection period. Dissections took approximately 1 hr to complete, and carcasses were kept on ice until fixation. Our HCR protocol was adapted from Molecular Instruments for use on larval carcasses and used Molecular Instruments reagents. In brief, after fixing in 4% formaldehyde, samples were permeabilized in detergent solution and pre-hybridized in 'Probe Hybridization Buffer' for 30 min at 37 °C. After pre-hybridization, samples were incubated in a probe solution consisting of 'Probe Hybridization Buffer' and probe sets overnight (12–24 hr) at 37 °C. Probe sets for any given gene were made against mRNA sequences conserved between all predicted gene isoforms. Following hybridization, samples were washed using 'Probe Wash Buffer' and SSCTween. Samples were then pre-incubated in 'Amplification Buffer' for 30 min. After pre-incubation samples were left in hairpin solution consisting of 'Amplification Buffer' and hairpins in the dark at room temperature overnight (12–24 hr). Samples were then washed, incubated with DAPI, and mounted using Invitrogen's Diamond Antifade Mountant. Slides were imaged using a confocal microscope and images were processed using Fiji. For image processing, a max-projection was generated of all images and the signal from each channel of the irradiated samples were auto-adjusted. These adjustments were propagated to all other wing-disc images of the same genotype.

## Acknowledgements

We are grateful to Nitya Mani for educating us on mathematical approaches to the study of heterogeneity, especially the Herfindahl-Hirschman Index. We thank Karthik Shekhar, Peter Sudmant, and Nicholas Everetts for their input on computational analyses. We thank Rebecca Chang for assistance with single-cell sample collection. We thank past and present members of the Hariharan Lab for their help dissecting larvae for scRNA-seq: Allie Cho, Nick Everetts, Sophia Friesen, Triny Ravindran, Danielle Spitzer, Melanie Worley, Ishita Srivastava, and Edgar Zepeda. We also thank Stephan Uwe Gerlach for his help with dissections. We thank Li Qingyang and Zohra Allata for their help dissecting for revision experiments. We thank Justin Choi for his preparation of 10x libraries. We thank Twinkle Bansal for proofreading the manuscript. We thank Gary Karpen for the p-H2Av antibody, Chris Doe for the Zfh2 antibody, the Developmental Studies Hybridoma Bank for antibodies, and the Bloomington Drosophila Stock Center for fly stocks, and FlyBase for hosting an invaluable compilation of information. This work was funded by an NIH grant R35 GM122490 (to IKH) and an NSF GRFP to JC.

# Additional information

## Funding

| Funder | Grant reference number | Author |
|---|---|---|
| National Institute of General Medical Sciences | R35 GM122490 | Iswar K Hariharan |
| National Science Foundation Graduate Research Fellowship Program | | Joyner Cruz |

The funders had no role in study design, data collection and interpretation, or the decision to submit the work for publication.

## Author contributions

Joyner Cruz, Conceptualization, Data curation, Formal analysis, Investigation, Visualization, Methodology, Writing – original draft, Writing – review and editing; William Y Sun, Data curation, Formal analysis, Investigation, Visualization, Writing – review and editing; Alexandra Verbeke, Formal analysis, Investigation, Visualization, Writing – review and editing; Iswar K Hariharan, Conceptualization, Supervision, Funding acquisition, Writing – original draft, Project administration, Writing – review and editing

## Author ORCIDs

Joyner Cruz ⓘ https://orcid.org/0009-0004-1076-4781
Alexandra Verbeke ⓘ https://orcid.org/0009-0002-3022-1755
Iswar K Hariharan ⓘ https://orcid.org/0000-0001-6505-0744

Reviewer #1 (Public review): https://doi.org/10.7554/eLife.106410.3.sa1
Reviewer #2 (Public review): https://doi.org/10.7554/eLife.106410.3.sa2
Reviewer #3 (Public review): https://doi.org/10.7554/eLife.106410.3.sa3
Author response https://doi.org/10.7554/eLife.106410.3.sa4

# Additional files

## Supplementary files

Supplementary file 1. Top markers for each cluster. Contains the following columns (Column; description): Subregion; cluster name Marker_Integrated_Global; Top marker gene when comparing cluster to all other cells in the integrated data. Pct_Diff_Integrated_Global; Percent of cells expressing top marker in cluster of interest minus percent of cells expressing in all other cells in the integrated data. Marker_R4K_Global; Top marker gene when comparing cluster to all other clusters in the 4000 rad condition. Pct_Diff_R4K_Global; Percent of cells expressing top marker in cluster of interest minus percent of cells expressing in all other cells in the 4000 rad condition. Marker_R0K_Global; Top marker gene when comparing cluster to all other clusters in the 0 rad condition. Pct_Diff_R0K_Global; Percent of cells expressing top marker in cluster of interest minus percent of cells expressing in all other cells in the 0 rad condition. Marker_Integrated_Region; Top marker gene when comparing cluster to all other clusters within its broad PD region in the integrated data. Pct_Diff_Integrated_Region; Percent of cells expressing top marker in cluster of interest minus percent of cells expressing in all other cells within its broad PD region in the integrated condition. Marker_R4K_Region; Top marker gene when comparing cluster to all other clusters within its broad PD region in the 4000 rad condition. Pct_Diff_R4K_Region; Percent of cells expressing top marker in cluster of interest minus percent of cells expressing in all other cells within its broad PD region in the 4000 rad condition. Marker_R0K_Region; Top marker gene when comparing cluster to all other clusters within its broad PD region in the 0 rad condition. Pct_Diff_R0K_Region; Percent of cells expressing top marker in cluster of interest minus percent of cells expressing in all other cells within its broad PD region in the 0 rad condition. Markers were only considered if (1) they were expressed in at least 10% of cells in the cluster being considered or its comparison group, (2) there was a minimum difference of 10%

between the cluster being considered or its comparison group, and (3) there was an enrichment of at least $log_2FC = 1$ in the cluster being considered. Ties were broken by taking the marker with the higher $log_2FC$.

Supplementary file 2. DEG 4000 rad vs 0 rad. Contains the following columns (Column; description): gene_name; Gene name. p_val; unadjusted p-value of Wilcoxon rank-sum test. avg_log2FC; average $log_2FC$ of all cells in 4000 rad condition versus all cells in 0 rad condition. pct.1; Percent of cells expressing given gene in 4000 rad condition. pct.2; Percent of cells expressing given gene in 0 rad condition. p_val_adj; Bonferroni corrected p-value from p_val column (corrected on total genes captured = 13,384). Genes were only included in this table if they were present in at least 1% of cells in either condition, had an adjusted P-value <0.05, and had an average $log_2FC$ of ≥*0.1*. This table was produced using the FindMarkers() function in Seurat v5.

Supplementary file 3. Apoptosis, DNA damage response (DDR), response to reactive oxygen species (ROS), cell cycle regulation, transcription factors (TFs), phosphatases, kinases, ligands, and receptors genes considered for HHI comparison (pre-filter). Contains the following columns (Column; description): genes_name; All genes captured in this data belonging to the considered categories, 1732 total. Category; Category each gene is classified into.

Supplementary file 4. Genes from *Supplementary file 3* with more than one category and their placements. Contains 81 genes that were found in more than one of the nine categories in *Supplementary file 3*. Contains the following columns (Column; description): genes_name; Genes from *Supplementary file 3* that were initially found in more than one category. Categories; Categories gene was found in. Category; Category each gene was chosen to be classified into (these are the classifications used in *Supplementary file 3*).

Supplementary file 5. Apoptosis, DNA damage response (DDR), response to reactive oxygen species (ROS), cell cycle regulation, transcription factors (TFs), phosphatases, kinases, ligands, and receptors genes considered for HHI comparison (post-filter, n=521). Contains the following columns (Column; description): gene_name; Gene name. Includes all 521 genes used in *Figures 3A and 8*. HHI_R0K; HHI score in the 0 rad condition. HHI_R4K; HHI score in the 4000 rad condition. p_val; p-value of Wilcoxon rank sum test for cluster of maximum FC. avg_log2FC; $Log_2FC$ between conditions of the cluster with the highest log2FC. pct.1; Percent of cells cluster of highest log2FC in the 4000 rad condition. pct.2; Percent of cells cluster of highest log2FC in the 0 rad condition. p_val_adj; Bonferroni adjusted p-value from p_val column (note only corrected for genes, not number of cluster FC tests ran). Cluster_max_FC; Cluster with the highest FC. Group; Gene category.

Supplementary file 6. Cell cycle gene markers for cell-cycle-based clusters. Contains the following columns (Column; description): gene; Gene name. avg_scaled_…; Average scaled expression in each cell cycle cluster (integrated). 6 columns. Percent_Expressed_…; Percentage of cells expressing given gene in each cell cycle cluster (integrated). 6 columns. This table includes all 175 genes used in cell-cycle-based clustering.

Supplementary file 7. Highly induced gene markers for cell-cycle-based clusters. This table includes the 359 genes that are highly induced after irradiation. These genes are expressed in at least 1% of cells in either condition, have an average $log_2FC$ ≥1, and an adjusted *P*-value <0.05. Contains the following columns (Column; description): gene; Gene name. avg_scaled_…; Average scaled expression in each cell cycle cluster in the 4000 rad condition. 6 columns. Percent_Expressed_…; Percentage of cells expressing given gene in each cell cycle cluster in the 4000 rad condition. 6 columns.

Supplementary file 8. Table containing FlyBase gene IDs, FlyBase gene CGs, and current gene symbols used in org.Dm.eg.db v3.18.0 R package to translate FlyBase IDs to Symbols. This table contains a list of 25,107 genes identified by their Flybase IDs, CGs, and current symbols at the time of analysis used to convert FlyBase IDs to Symbols. Note, not all genes were captured in this dataset. Contains the following columns (Column; description): FLYBASE; FlyBase ID. FLYBASECG; FlyBase CG. SYMBOL; Symbol.

MDAR checklist

## Data availability

Raw scRNA-seq files can be accessed at GEO, accession number GSE285361: https://www.ncbi.nlm.nih.gov/geo/query/acc.cgi?acc=GSE285361 Example code used to generate HHI and Euclidean expression distance scores can be found at: https://github.com/HariharanLab/Cruz_Sun_Verbeke_Hariharan (copy archived at *Jojace, 2025*).

The following dataset was generated:

| Author(s) | Year | Dataset title | Dataset URL | Database and Identifier |
|---|---|---|---|---|
| Cruz J, Sun WY, Verbeke A, Hariharan IK | 2024 | Single of X-ray irradiated Drosophila wing discs reveals heterogeneity related to cell-cycle status and cell location-cell transcriptomics | https://www.ncbi.nlm.nih.gov/geo/query/acc.cgi?acc=GSE285361 | NCBI Gene Expression Omnibus, GSE285361 |

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
