## [Editor Report · eLife Assessment]

This **important** study uses standard single-cell RNA-seq analyses combined with methods from the social sciences to reduce heterogeneity in gene expression in Drosophila imaginal wing disc cells treated with 4000 rads of ionizing radiation. The use of this methodology from social sciences is novel in Drosophila and allows them to identify a subpopulation of cells that is disproportionately responsible for much of the radiation-induced gene expression. Their **compelling** analyses reveal genes that are expressed regionally after irradiation, including ligands and transcription factors that have been associated with regeneration, as well as others whose roles in response to irradiation are unknown. This paper would be of interest to researchers in the field of DNA damage responses, regeneration, and development.

---

## [Referee Report · Reviewer #1 (Public review)]

Summary:

The authors analyze transcription in single cells before and after 4000 rads of ionizing radiation. They use Seuratv5 for their analyses, which allows them to show that most of the genes cluster along the proximal-distal axis. Due to the high heterogeneity in the transcripts, they use the Herfindahl-Hirschman index (HHI) from Economics, which measures market concentration. Using the HHI, they find that genes involved in several processes (like cell death, response to ROS, DNA damage response (DDR)) are relatively similar across clusters. However, ligands activating the JAK/STAT, Pvr, and JNK pathways and transcription factors Ets21C and dysf are upregulated regionally. The JAK/STAT ligands Upd1,2,3 require p53 for their upregulation after irradiation, but the normal expression of Upd1 in unirradiated discs is p53-independent. This analysis also identified a cluster of cells that expressed tribbles, encoding a factor that downregulates mitosis-promoting String and Twine, that appears to be G2/M arrested and expressed numerous genes involved in apoptosis, DDR, the aforementioned ligands and TFs. As such, the tribbles-high cluster contains much of the heterogeneity.

Strengths:

(1) The authors have used robust methods for rearing Drosophila larvae, irradiating wing discs and analyzing the data with Seurat v5 and HHI.

(2) These data will be informative for the field.

(3) Most of the data is well-presented.

(4) The literature is appropriately cited.

Weaknesses

The authors have addressed my concerns in the revised article.

---

## [Referee Report · Reviewer #2 (Public review)]

This manuscript investigates the question of cellular heterogeneity using the response of Drosophila wing imaginal discs to ionizing radiation as a model system. A key advance here is the focus on quantitatively expressing various measures of heterogeneity, leveraging single-cell RNAseq approaches. To achieve this goal, the manuscript creatively uses a metric from the social sciences called the HHI to quantify the spatial heterogeneity of expression of individual genes across the identified cell clusters. Inter- and intra-regional levels of heterogeneity are revealed. Some highlights include identification of spatial heterogeneity in expression of ligands and transcription factors after IR. Expression of some of these genes shows dependence on p53. An intriguing finding, made possible by using an alternative clustering method focusing on cell cycle progression, was the identification of a high-trbl subset of cells characterized by concordant expression of multiple apoptosis, DNA damage repair, ROS related genes, certain ligands and transcription factors, collectively representing HIX genes. This high-trbl set of cells may correspond to an IR-induced G2/M arrested cell state.

Overall, the data presented in the manuscript are of high quality but are largely descriptive. This study is therefore perceived as a resource that can serve as an inspiration for the field to carry out follow-up experiments.

The authors responded well to my suggestions for improvement, which were incorporated in the revised version of the manuscript.

---

## [Referee Report · Reviewer #3 (Public review)]

Summary:

Cruz and colleagues report a single cell RNA sequencing analysis of irradiated Drosophila larval wing discs. This is a pioneering study because prior analyses used bulk RNAseq analysis so differences at single cell resolution were not discernable. To quantify heterogeneity in gene expression, the authors make clever use of a metric used to study market concentration, the Herfindahl-Hirschman Index. They make several important observations including region-specific gene expression coupled with heterogeneity within each region and the identification of a cell population (high Trbl) that seems disproportionately responsible for radiation-induced gene expression.

Strengths:

Overall, the manuscript makes a compelling case for heterogeneity in gene expression changes that occurs in response to uniform induction of damage by X-rays in a single layer epithelium. This is an important finding that would be of interest to researchers in the field of DNA damage responses, regeneration and development.

Weaknesses:

The authors have addressed my concerns adequately with changes made in the revised version.

---

## [Author Response]

The following is the authors’ response to the original reviews

**Reviewing Editor Comment:**
The reviewers felt that the study could be improved by (1) better integrating the results with the existing literature in the field

(1) In the Introduction and Results section of the manuscript, we had made every attempt to cite the relevant literature. (Reviewer 1 stated that “The literature is appropriately cited”). We agree with the Reviewing Editor that rather than simply cite the relevant literature, we could have done a better job of integrating our findings with what has been previously discovered by others. We have attempted to do this in the revised manuscript. Also, we have included many additional citations in the Introduction and in the first section of the Results where work by others has provided a framework for interpreting our single-cell studies.

and (2) manipulating Trib expression and analyzing the expression of 1-2 HIX genes.

(2) We are grateful for this suggestion. As suggested by the Reviewing Editor we have attempted to increase and decrease *trbl* expression and assess the effect on expression of two genes, *Swim* and CG15784.

We increased *trbl* levels in the wing pouch using *rn-Gal4*, *tub-Gal80ts* and *UAS-trbl*. By transferring larvae for 24 h from 18oC to 31oC, we were able to induce *trbl* expression in the wing pouch. When these larvae were irradiated at 4000 rad, we found reduced levels of apoptosis in the wing pouch of discs that overexpressed *trbl* (Figure 7-figure supplement 1). This indicated that upregulation of *trbl* is radioprotective. Consistent with our findings, others have previously shown that upregulation of *trbl* and stalling in the G2 phase of the cells cycle protects cells from JNK-induced apoptosis (Cosolo et al., 2019, PMID:30735120) or that downregulating the G2/M progression promoting factor *string* protects cells from X-ray radiation induced apoptosis (Ruiz-Losada et al., 2021, PMID:34824391).

As suggested by the Reviewing Editor, we also examined the effect of *trbl* overexpression on the induction of two “highly induced by X-ray irradiation (HIX)” gene, *Swim* and CG15784. Increasing *trbl* expression had no effect on the induction of *Swim* and only a modest decrease in the induction of CG15784 (Figure 7-figure supplement 2). Thus, increasing *trbl* expression, is in itself, insufficient to promote HIX gene expression indicating that other factors are necessary for HIX gene induction.

We also attempted to reduce *trbl* expression, using three different RNAi lines. While some of these lines have been used previously by others to reduce *trbl* expression under unirradiated conditions (Cosolo et al., 2019, PMID:30735120), we nevertheless wanted to check if they reduced *trbl* induction following irradiation. For each of the three lines, we observed no obvious reduction in *trbl* RNA following irradiation when visualized using HCR (Author response image 1). Thus, any effects on gene expression that we observe could not be attributed to a decrease in *trbl* expression. We have therefore included the images showing a lack of knockdown in this Response to Reviews document but not included these experiments in the revised manuscript.

**Author response image 1. sa4fig1:** RNA in situ hybridizations using the hybridization chain reaction performed using probes to *trbl*. In A-F, the RNAi is expressed using *nubbin-Gal4*. In G-I the RNAi is expressed using *rn-Gal4*, *tub-Gal80ts*. *white*-RNAi was used as a control (A, B, G, H). Three different RNAi lines directed against *trbl* were tested: Vienna lines VDRC 106774 (C, D) and VDRC 22113 (E, F), and Bloomington line BL42523. In no case was a reduction in *trbl* RNA upregulation in the wing pouch following 4000 rad observed, except for one disc (n = 6) of VDRC 106774 crossed to *nubbin-gal4.*

**Reviewer #1 (Public review):**
Summary:The authors analyze transcription in single cells before and after 4000 rads of ionizing radiation. They use Seuratv5 for their analyses, which allows them to show that most of the genes cluster along the proximal-distal axis. Due to the high heterogeneity in the transcripts, they use the Herfindahl-Hirschman index (HHI) from Economics, which measures market concentration. Using the HHI, they find that genes involved in several processes (like cell death, response to ROS, DNA damage response (DDR)) are relatively similar across clusters. However, ligands activating the JAK/STAT, Pvr, and JNK pathways and transcription factors Ets21C and dysf are upregulated regionally. The JAK/STAT ligands Upd1,2,3 require p53 for their upregulation after irradiation, but the normal expression of Upd1 in unirradiated discs is p53-independent. This analysis also identified a cluster of cells that expressed tribbles, encoding a factor that downregulates mitosis-promoting String and Twine, that appears to be G2/M arrested and expressed numerous genes involved in apoptosis, DDR, the aforementioned ligands, and TFs. As such, the tribbles-high cluster contains much of the heterogeneity.Strengths:(1) The authors have used robust methods for rearing Drosophila larvae, irradiating wing discs, and analyzing the data with Seurat v5 and HHI.(2) These data will be informative for the field.(3) Most of the data is well-presented(4) The literature is appropriately cited.

We thank the reviewer for these comments.

Weaknesses:(1) The data in Figure 1 are single-image representations. I assume that counting the number of nuclei that are positive for these markers is difficult, but it would be good to get a sense of how representative these images are and how many discs were analyzed for each condition in B-M.

For each condition at least 5 discs were imaged but we imaged up to 15 discs in some cases. We tried to choose a representative disc for each condition after looking at all of them. All discs imaged under each condition are shown below; the disc chosen for the figure is indicated with an asterisk. All scale bars are 100 mm.

**Author response image 2. sa4fig2:** Images for discs shown in Manuscript Figure 1panels B, C.

**Author response image 3. sa4fig3:** Images for discs shown in Manuscript Figure 1panels D, E.

**Author response image 4. sa4fig4:** Images used in Manuscript Figure 1, F, G.

**Author response image 5. sa4fig5:** Images used in Manuscript Figure 1H, I.

**Author response image 6. sa4fig6:** Images used in Manuscript Figure 1J, K.

**Author response image 7. sa4fig7:** Images used in Manuscript Figure 1L, M.

(2) Some of the figures are unclear.

It is unclear to us exactly which figures the Reviewer is referring to. Perhaps this is the same issue mentioned below in “Recommendations for the authors”. We address it below.

**Reviewer #1 (Recommendations for the authors):**
(1) Regarding Figure 1, what is stained in blue? Is it DAPI? If so, this should be added to the figure legend.

Thank you for pointing out this omission. This has been addressed in the revised manuscript.

It is very difficult to see blue on black, so could the authors please outline the discs?Alternatively, they could show DAPI in green and the markers (pH2Av, etc) in magenta.

We used DAPI (blue) as a way of outlining the discs. While we appreciate the reviewer’s concern, after reviewing the images, we found that the blue is clearly visible when the document is viewed on the screen. It is less obvious if the document is printed on some kinds or printers. Since boosting this channel would make the signal from the channels more difficult to see, we left the images as they were.

(2) Figure 3, Figure Supplement 2, panel B. It is not possible to read the gene names in the panel's current form. Please break this up into 4 lines (as much as possible from the current 2).

Thank you for this suggestion. We have done this in the revised manuscript.

**Reviewer #2 (Public review):**
This manuscript investigates the question of cellular heterogeneity using the response of Drosophila wing imaginal discs to ionizing radiation as a model system. A key advance here is the focus on quantitatively expressing various measures of heterogeneity, leveraging single-cell RNAseq approaches. To achieve this goal, the manuscript creatively uses a metric from the social sciences called the HHI to quantify the spatial heterogeneity of expression of individual genes across the identified cell clusters. Inter- and intra-regional levels of heterogeneity are revealed. Some highlights include the identification of spatial heterogeneity in the expression of ligands and transcription factors after IR. Expression of some of these genes shows dependence on p53. An intriguing finding, made possible by using an alternative clustering method focusing on cell cycle progression, was the identification of a high-trbl subset of cells characterized by concordant expression of multiple apoptosis, DNA damage repair, ROS-related genes, certain ligands, and transcription factors, collectively representing HIX genes. This high-trbl set of cells may correspond to an IR-induced G2/M arrested cell state.Overall, the data presented in the manuscript are of high quality but are largely descriptive. This study is therefore perceived as a resource that can serve as an inspiration for the field to carry out follow-up experiments.

Thank you for your assessment of the work.

**Reviewer #2 (Recommendations for the authors):**
I suggest two major points for improvement:(1) It is important to test whether manipulation of trbl levels (i.e., overexpression, knockdown, mutation) would result in measurable biological outcomes after IR, such as altered HIX gene expression, altered cell cycle progression, or both. This may help disentangle the question of whether high trbl expression and correlated HIX gene expression are a cause or consequence of G2/M stalling.

We have described these experiments at the beginning of this Response to Reviews document when addressing the comments made by the Reviewing Editor. Please see Figure 7, figure supplements 1 and 2. These experiments suggest that upregulation of *trbl* offers some protection from radiation-induced death, yet it is itself insufficient to induce expression of two HIX genes tested. As we have also described earlier, three different RNAi lines tested did not reduce *trbl* upregulation after irradiation.

(2) A more extensive characterization of the high-trbl cell state would also be appropriate, particularly in terms of their relationship to the cell cycle.

We attempted to address this issue in two ways. First, we used the expression of a *trbl-gfp* transgene and RNA in-situ hybridization experiments to visualize the distribution of the high-*trbl* cells (shown in new manuscript figure, Figure 6-figure supplement 3). When examining *trbl* RNA in irradiated discs, there is no obvious demarcation between cells that express high levels of *trbl* and other cells. This is also apparent in the UMAP shown in Figure 6A and A’. Most cells seem to express *trbl*; cells in the “high *trbl*” cluster simply express more *trbl* than others. We observed cells expressing *trbl* and *PCNA* as well as cells expressing only one of those two genes at detectable levels. Thus, it was not possible to distinguish the “high *trbl*” cells from other cells by this approach.

We decided instead to focus on examining the expression of other cell-cycle genes in the high-*trbl* cluster. We have added a paragraph in the Results section that details our findings. Many transcriptional changes are indeed consistent with stalling in G2 such as high levels of *trbl* and low levels of *string* (stg). Additionally, that the cells are likely in G2 is consistent with reduced levels of genes that are normally expressed at other stages of the cell cycle: G1 genes such as *E2f1* and *Dp,* S-phase genes such as several Mcm genes, *PCNA* and *RnrS,* and genes that encode mitotic proteins such as *polo*, *Incenp* and *claspin*. There are however, several anomalies such as slightly increased expression of the early-G1 cyclin, *CycD*, and the retinoblastoma ortholog *Rbf*. Thus, at least as assessed by the transcriptome, this cluster may not correspond to a cell state that is found under normal physiological conditions.

(3) Minor: p. 12, line 3. Figure 5A is mentioned, but it seems that it should be 4A instead.

Thank you for pointing this out. We have addressed this in our revisions.

**Reviewer #3 (Public review):**
Strengths:Overall, the manuscript makes a compelling case for heterogeneity in gene expression changes that occur in response to uniform induction of damage by X-rays in a single-layer epithelium. This is an important finding that would be of interest to researchers in the field of DNA damage responses, regeneration, and development.Weaknesses:This work would be more useful to the field if the authors could provide a more comprehensive discussion of both the impact and the limitations of their findings, as explained below.Propidium iodide staining was used as a quality control step to exclude cells with a compromised cell membrane. But this would exclude dead/dying cells that result from irradiation. What fraction of the total do these cells represent? Based on the literature, including works cited by the authors, up to 85% of cells die at 4000R, but this likely happens over a longer period than 4 hours after irradiation. Even if only half of the 85% are PI-positive by 4 hr, this still removes about 40% of the cell population from analysis. The remaining cells that manage to stay alive (excluding PI) at 4 hours and included in the analysis may or may not be representative of the whole disc. More relevant time points that anticipate apoptosis at 4 hr may be 2 hr after irradiation, at which time pro-apoptotic gene expression peaks (Wichmann 2006). Can the authors rule out the possibility that there is heterogeneity in apoptosis gene expression, but cells with higher expression are dead by 4 hours, and what is left behind (and analyzed in this study) may be the ones with more uniform, lower expression? I am not asking the authors to redo the study with a shorter time point, but to incorporate the known schedule of events into their data interpretation.

We thank the reviewer for these important comments. The generation of single-cell RNA-seq data from irradiated cells is tricky. Many cells have already died. Even those that do not incorporate propidium iodide are likely in early stages of apoptosis or are physiologically unhealthy and likely made it through our FACS filters. Indeed, in irradiated samples up to 57% of sequenced cells were not included in our analysis since their RNA content seemed to be of low quality. It is therefore likely that our data are biased towards cells that are less damaged. As advised by the reviewer, we will include a clearer discussion of these issues as well as the time course of events and how our analysis captures RNA levels only at a single time point.

If cluster 3 is G1/S, cluster 5 is late S/G2, and cluster 4 is G2/M, what are clusters 0, 1, and 2 that collectively account for more than half of the cells in the wing disc? Are the proportions of clusters 3, 4, and 5 in agreement with prior studies that used FACS to quantify wing disc cells according to cell cycle stage?

Work by others (Ruiz-Losada et al., 2021, PMID:34824391) has shown that almost 80% of cells have a 4C DNA content 4 h after 4,000 rad X-ray irradiation. The high-*trbl* cluster accounts for only 18% of cells and can therefore account for a minority of cells with a 4C DNA content.

Thus clusters 0, 1 and 2 could potentially contain other populations that also have a 4C DNA content. Importantly, similar proportions of cells in these clusters are also observed in unirradiated discs.

We expect that clusters 1 and 2 are largely comprised of cells in G2/M. Together, these clusters are marked by some genes previously found to be higher in FACS separated G2 cells compared to G1 cells (Liang et al., 2014, PMID: 24684830). These genes include *Det, aurA,* and *ana1*. Strangely, cluster 0 is not strongly marked by any of the 175 cell cycle genes used in our clustering (*eff* being the strongest marker) and has a lower-than-average expression of 165/175 cell cycle genes. Cluster 0 is however marked by the genes *ac* and *sc*, which are known to be expressed in proneuronal cell clusters interspersed throughout the disc that stall in G2 and form mitotically quiescent domains (Usui & Kimura 1992, Development, 116 (1992), pp. 601-610 (no PMID); Nègre et al., 2003, PMID: 12559497). Given these observations, we hypothesize that cluster 0 is largely comprised of stalled G2 cells like those found in ac/sc-expressing proneural clusters.

The EdU data in Figure 1 is very interesting, especially the persistence in the hinge. The authors speculate that this may be due to cells staying in S phase or performing a higher level of repair-related DNA synthesis. If so, wouldn't you expect 'High PCNA' cells to overlap with the hinge clusters in Figures 6G-G'? Again, no new experiments are needed. Just a more thorough discussion of the data.

We have found that the locations of elevated PCNA expression do not always correlate with the location of EdU incorporation either by examining scRNA-seq data or by using HCR to detect PCNA. PCNA expression is far more widespread as we now show in Figure 6-figure supplement 3.

Trbl/G2/M cluster shows Ets21C induction, while the pattern of Ets21C induction as detected by HCR in Figures 5H-I appears in localized clusters. I thought G2/M cells are not spatially confined. Are Ets21C+ cells in Figure 5 in G2/M? Can the overlap be confirmed, for example, by co-staining for Trbl or a G2/M marker with Ets21C?

The data show that the high-*trbl* cells are higher in *Ets21C* transcripts relative to other cell-cycle-based clusters after irradiation. This does not imply that high-*trbl*-cells in all regions of the disc upregulate *Ets21C* equally. *Ets21C* expression is likely heterogeneous in both ways – by location in the disc and by cell-cycle state.

Induction of dysf in some but not all discs is interesting. What were the proportions? Any possibility of a sex-linked induction that can be addressed by separating male and female larvae?

We can separate the cells in our dataset into male and female cells by expression of *lncRNA:roX1/2.* When we do this, we see X-ray induced *dysf* expressed similarly in both male and female cells. We think that it is therefore unlikely that this difference in expression can be attributed to cell sex. Another possibility is that *dysf* upregulation might be acutely sensitive to the developmental stage of the disc. This would require experiments with very precisely-staged larvae. We have not investigated this further as it is not a central issue in our paper.

**Reviewer #3 (Recommendations for the authors):**
Please check the color-coding in Figure 1A. The region marked as pouch appears to include hinge folds that express Zfh2 (a hinge marker) in Figure 2A (even after accounting for low Zfh2 expression in part of the pouch).

We have corrected this and have marked the pouch region based on the analysis of expression of different hinge and pouch markers by Ayala-Camargo et al. 2013 (PMID 2398534).

The statement 'Furthermore, within tissues, stem cells are most sensitive while differentiated cells are relatively radioresistant' needs to be qualified, as there are differences in radiosensitivity of adult versus embryonic stem cells (e.g., PMID: 30588339)

We thank the reviewer for bringing this point to our attention and for pointing us to an article that addresses this issue in detail. We appreciate that our statement was rather simplistic – we have modified it and added two additional references.